# TBK1 restricts IRGQ-mediated autophagy

Uxia Gestal-Mato [1], Pauline Lascaux [1], Sergio Alejandro Poveda-Cuevas [1,2], Alberto Cristiani [1,2], Belinda Camp [3], Mahyar Aghapour [3], Aparna Viswanathan Ammanath [1], Ramachandra M. Bhaskara [1,2,4], Ivan Dikic [1,2,4,5] ✉ & Lina Herhaus [1,3] ✉

The autophagy-lysosome system directs the degradation of a wide variety of cytoplasmic cargo such as damaged organelles, protein aggregates, and invading pathogens. The autophagy receptor IRGQ harbors two distinct LIR domains, with LIR1 exhibiting high selectivity for GABARAPL2. Proteomic, biochemical, and high-throughput microscopy studies reveal that the IRGQ-GABARAPL2 complex functions as a hub for the interaction between hATG8s and the autophagy initiation machinery, promoting their lipidation and overall autophagic flux. The interaction of IRGQ with GABARAPL2 is regulated via TBK1. Upon TBK1 activation, GABARAPL2 is phosphorylated on S10, which disrupts IRGQ-GABARAPL2 complexation and therefore its interaction with the autophagy initiation machinery, resulting in a reduction of the autophagic flux of GABARAPL2 and IRGQ-cargo, without affecting bulk autophagy. These findings broaden IRGQ's role in autophagy, identifying it as an interaction hub for autophagy initiation that is negatively regulated by TBK1.

Autophagy is a highly conserved, catabolic process that secures cellular homeostasis by recycling critical metabolites during starvation. Dysregulation of autophagy has been widely implicated in various pathophysiological processes such as aging, cancer, metabolic and neurodegenerative disorders, as well as cardiovascular and pulmonary diseases[1,2]. During the process of autophagy, a de-novo double-membraned vesicle termed the phagophore forms around demarcated cargo or bulk cytosol that will be degraded. The phagophore membrane arises from ATG9-positive vesicles, which serve as the initial membrane seed. ATG2-mediated lipid transport from the endoplasmic reticulum (ER) is required for expansion and subsequent phagophore formation[3,4] and is governed by a multiprotein complex composed of ULK1/2, ATG13, FIP200, and ATG101[5–8]. Additionally, there are two systems involving ubiquitin-like proteins that contribute to the expansion of the phagophore and its subsequent fusion with the lysosome: the ATG12 conjugation pathway and the processing of the human ATG8 family (hATG8) by ATG7 and ATG3[9].

The hATG8s are ubiquitin-like proteins that can be divided into two subfamilies: the LC3s (LC3A, LC3B, LC3C) and the GABARAPs (GABARAP, GABARAPL1, GABARAPL2). All members of this family are conjugated to phosphatidylethanolamine (PE) in the membrane of the growing phagophore (referred to as lipidation) by ATG7 (an E1-like enzyme), ATG3 (an E2-like enzyme), and the ATG12-ATG5-ATG16L1 complex (an E3-like enzyme)[10]. A prerequisite for PE conjugation is processing of pro-LC3 by the ATG4 protease. Closure of the phagophore results in the completed autophagosome that can fuse with lysosomes containing hydrolytic enzymes.

Based upon cargo being selectively delivered for degradation, autophagy has been classified into numerous subtypes, such as mitophagy, ER-phagy, or xenophagy[11]. Cargo selection in these pathways is often achieved via the binding of an autophagy receptor that harbors a consensus LC3-interacting region (LIR), a stretch of 4 amino acids to facilitate its direct binding to the LDS (LIR-docking site) of LC3 orthologs.

Autophagy receptors play a central role in linking specific cargo to the core autophagy machinery[12]. Thus, receptors serve as molecular adapters that ensure selective cargo recognition and efficient sequestration within autophagosomes. In addition, some of them act

[1]Institute of Biochemistry II, Medical Faculty, Goethe University Frankfurt, Frankfurt am Main, Germany. [2]Buchmann Institute for Molecular Life Sciences, Riedberg Campus, Goethe University Frankfurt, Frankfurt am Main, Germany. [3]Helmholtz-Zentrum für Infektionsforschung GmbH, Braunschweig, Germany. [4]IMPRS on Cellular Biophysics, Frankfurt am Main, Germany. [5]Max Planck Institute of Biophysics, Goethe University Frankfurt, Frankfurt am Main, Germany. ✉e-mail: dikic@biochem2.uni-frankfurt.de; lina.herhaus@helmholtz-hzi.de

as autophagy initiation hubs and can nucleate autophagosome formation by directly recruiting components of the core autophagy machinery, such as the ULK1 complex or ATG proteins, to the cargo site[13–15]. Receptors like NDP52 and OPTN have been demonstrated to interact with both cargo and core machinery, effectively coordinating cargo selection with autophagosome biogenesis. This dual role highlights how receptors actively shape the specificity and efficiency of selective autophagy responses, revealing them as dynamic organizers of selective autophagy, capable of modulating cargo degradation based on cellular context and signaling inputs.

We recently identified IRGQ as an emerging receptor impacting the quality control of MHC class I (HLA) molecules[16]. IRGQ interacts with GABARAPL2 and LC3B via two distinct conserved LIRs and is trafficked to lysosomes in an autophagy-dependent manner. The Tank1 binding kinase (TBK1) is a critical regulator of various forms of autophagy via the phosphorylation of autophagy receptors, promoting autophagic flux[17]. Additionally, our previous work demonstrated that TBK1 can also regulate LC3 orthologs to control autophagic flux[18]. Here, we present IRGQ as a hub for autophagy initiation that is restricted by TBK1. Through its unique dual LIR-binding, IRGQ serves

as an interaction hub for the hATG8s and the autophagy initiation machinery, promoting their lipidation and flux. TBK1 activation results in phosphorylation of GABARAPL2 S10, causing the disruption of the hub and impairing the autophagic flux of LC3B, GABARAPL2, and IRGQ's cargo molecule, HLA. All together, we uncover TBK1 as a negative regulator of IRGQ's autophagic axis, showcasing the refinement of TBK1-regulated autophagic responses.

## Results

### IRGQ facilitates the interaction of hATG8s and the autophagy initiation machinery

Autophagy receptors harbor LIR domains crucial for interaction with hATG8s. IRGQ has two LIRs in its sequence, AlphaFold3 prediction shows LIR1 (aa186-189) facing the G-protein fold of its structure and LIR2 (aa421-424) in proximity to a disordered loop region (Fig. 1A). AlphaFold2-multimer modeling performed in a previous study[16], suggested that out of the 6 hATG8s, GABARAPL2 was strongly preferred for the LIR1-LDS binding mode with 72% of the top models resulting in such complex (Figs. 1B and S1C). Although there was high occupancy by LC3B for the LIR2-LDS models, there was no clear preference

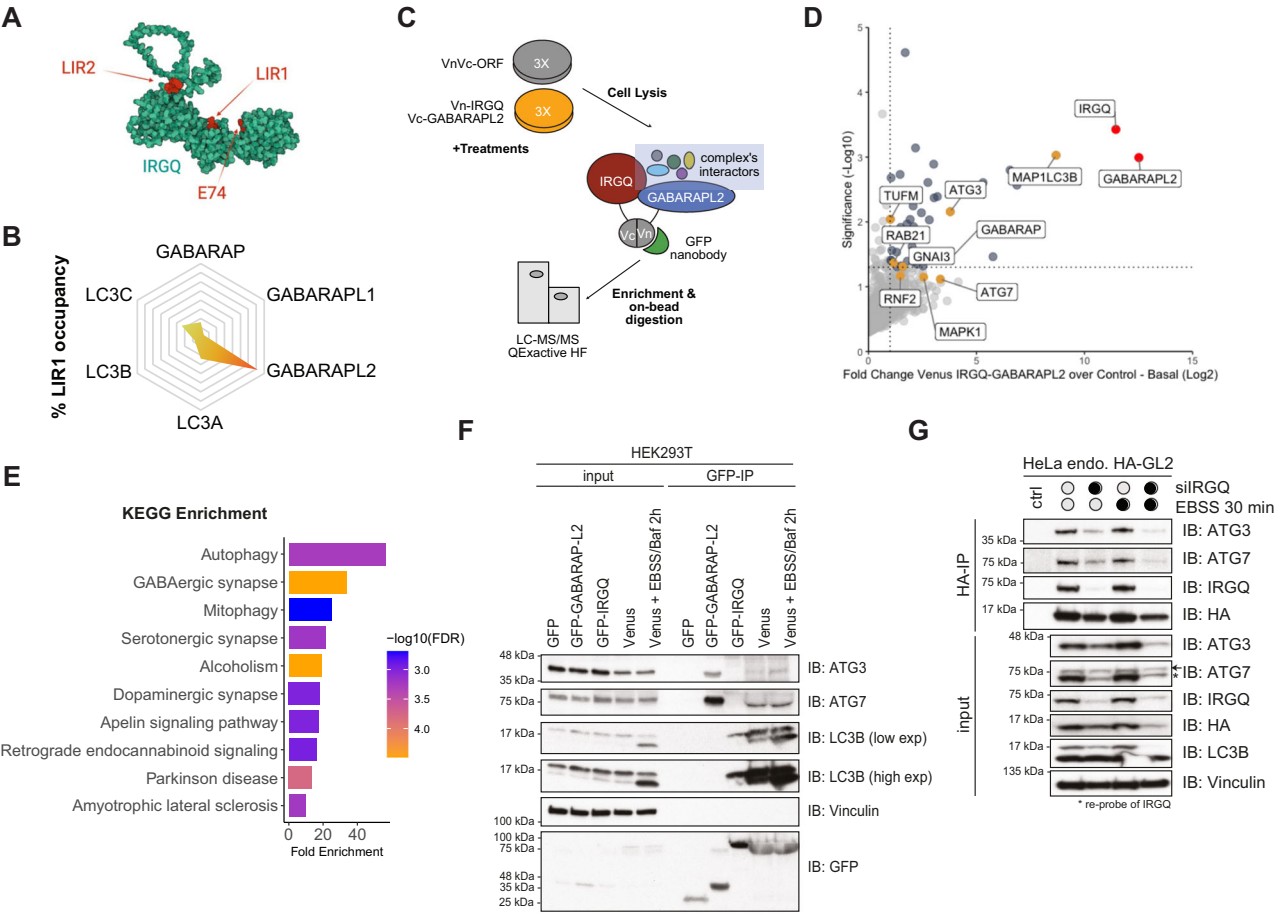

**Fig. 1 | IRGQ facilitates the interaction of hATG8s and the autophagy initiation machinery. A** IRGQ AlphaFold3 structure prediction highlighting interaction interfaces to hATG8s in red (E74R, LIR1, and LIR2). **B** Percentage of LIR1-LDS binding mode as the predicted complex between IRGQ-hATG8 in the 25 top-ranked AlphaFold2-multimer models from previous study[16]. **C** Experimental set-up for label free interactome studies of the IRGQ-GABARAPL2 complex. HEK293T cells were transfected with VnVc-ORF or Vn-IRGQ-Vc-GABARAPL2, lysates used for GFP-IPs and processed for mass spectrometry. Data was analyzed with MaxQuant and Perseus; n = 3. **D** Volcano-plot representing the Student's T-test difference from Vn-IRGQ-Vc-GABARAPL2 over VnVc-ORF IPs and −Log Student's two-sided T-test p-value from Vn-IRGQ-Vc-GABARAPL2 over VnVc-ORF IPs in untreated conditions.

The baits IRGQ and GABARAPL2 are marked in red, significant interaction partners in blue and autophagy-related proteins marked in yellow. **E** KEGG Enrichment Analysis of all significant interactors from Vn-IRGQ-Vc-GABARAPL2 over VnVc-ORF IPs in Basal conditions performed with ShinyGO 0.77. **F** SDS-PAGE and Western blot of GFP-IPs from HEK293T transfected with GFP-empty, GFP-GABARAPL2, GFP-IRGQ or Venus (Vn-IRGQ-Vc-GABARAPL2). Representative image of two independent experiments. **G** SDS-PAGE and Western blot of HA-IPs from endogenously tagged HA-GABARAPL2 cells upon treatment with IRGQ siRNA and/or EBSS starvation for 30 min. Representative image of two independent experiments. Source data are provided as a Source data file.

towards a specific hATG8 (Fig. S1A). The presence of two distinct LIR domains in the autophagy receptor IRGQ with one LIR motif having such specificity towards a single hATG8 is not known for any other autophagy receptor. To further study this phenomenon, we compared the N-terminal sequences of all hATG8s since the α−2 of GABARAPL2 is key for the interaction interface with IRGQ[16]. Sequence comparison shows that GABARAPL2 is the only hATG8 able to fit in the LIR1-G1 loop pocket of IRGQ, due to the presence of bulky residues in the other members of the family that would impede binding (Fig. S1B). GST pulldowns confirmed that the N-terminal region of GABARAPL2 is crucial for interaction with IRGQ (Fig. S1D).

To further dissect this striking evolutionary specificity, we queried the interactome of the IRGQ-GABARAPL2 complex with proteomics by using the Venus complementation affinity purification (BiCAP) method (Fig. 1C), which forms both in basal and autophagy-induced conditions (Fig. S1E). Immunoprecipitations using GFP-trap nanobodies in HEK293T cells revealed that LC3B, GABARAP, ATG3 and ATG7 are enriched in Venus-IPs versus control vectors, and that the binding to GABARAP and LC3B is further increased upon autophagy induction (Figs. 1D and S1F). KEGG Pathway Enrichment shows that interactors of the complex belong to pathways related to autophagy and membrane protein trafficking (Fig. 1E). These results were confirmed with co-immunoprecipitation using the Venus expressing cell lines, where GFP-GABARAPL2 binds endogenous ATG3 and ATG7, while GFP-IRGQ binds endogenous LC3B (Fig. 1F). Interestingly, when IRGQ and GABARAPL2 form a complex (Venus), they bind all three proteins (Fig. 1F). Conversely, the depletion of IRGQ from cells decreases the binding of endogenous GABARAPL2 to endogenous ATG3 and ATG7 (Fig. 1G). Due to both LIR motifs, IRGQ affects the co-precipitation of GABARAPL2, LC3B, and the autophagy initiation machinery. Taken together, the IRGQ-GABARAPL2 complex enables the recruitment of autophagy machinery and LC3B, acting as an autophagic hub.

## The IRGQ-GABARAPL2 autophagy hub promotes hATG8 lipidation

ATG3 (an E2-like enzyme) and ATG7 (an E1-like enzyme) mediate the conjugation of ATG8 proteins to phosphatidylethanolamine (PE), a key step in the autophagy pathway[10]. Since IRGQ affects the interaction between the hATG8s and the autophagy initiation machinery, we tested whether IRGQ assists in GABARAPL2 or LC3B lipidation. The overexpression of WT IRGQ results in increased LC3B (Figs. 2A and S2A) and GABARAPL2 lipidation (Fig. 2B). IRGQ mutants targeting the N-terminal region that mediates interaction with GABARAPL2 fail to promote GABARAPL2 lipidation despite comparable expression levels, demonstrating that lipidation depends on the integrity of the IRGQ-GABARAPL2 interaction rather than on IRGQ overexpression alone. Similarly, cells expressing the IRGQ-GABARAPL2 complex (Venus) exhibit increased LC3B lipidation at early time points of EBSS induction, compared to empty vector (Figs. 2C and S2B). Interestingly, overexpressing GABARAPL2 alone seemingly increases the unlipidated LC3B, possibly due to a sequestration of ATG3 and ATG7 by GABARAPL2 (Fig. 2C).

Upon autophagy induction, hATG8s are lipidated and decorate autophagosomal membranes. In particular, LC3-II foci formation is often used as a measurement for autophagic flux[19,20]. Thus, we used immunofluorescence studies to monitor the effect of the IRGQ-GABARAPL2 complex on LC3B lipidation. Confocal images of cells expressing the IRGQ-GABARAPL2 Venus complex suggested that expressing-cells had a higher level of LC3B staining intensity and puncta formation in comparison to non-expressing cells, highlighted in white (Fig. 2D). Image quantification confirmed this observation, as Venus intensity positively correlated with LC3B intensity and more LC3B foci were present in those cells where the levels of the IRGQ-GABARAPL2 complex was higher (Fig. 2E). Analysis of LAMP1 staining reveals similar results regarding the number of foci, although intensity

does not vary as much in relation to Venus expression, compared to LC3B (Fig. S2C). To directly assess autophagy flux and exclude the possibility that increased LC3B lipidation and foci reflect a block in autophagy, we analyzed p62 turnover by immunoblotting (Fig. S2D). As expected, p62 levels were reduced upon EBSS treatment, confirming active autophagy flux. Importantly, upon Venus-based co-over-expression of IRGQ and GABARAPL2, p62 degradation was further enhanced, indicating increased autophagic flux rather than impaired turnover.

## IRGQ-GABARAPL2 complex stability is disrupted upon GABARAPL2 S10D mutation

The interaction interface between the two proteins shows E74 of IRGQ, a key residue for the complex interaction[16], bridging to S10 of GABARAPL2 (Fig. 3A). Notably, this residue is heavily conserved among species (Fig. S3A). Our previous research showed that TBK1 phosphorylates members of the ATG8 family on several residues, GABARAPL2 S10 among them[18]. Mutational analysis in a TBK1 in vitro kinase assay reveals that S10 is the major TBK1-mediated phosphorylation site, since the S10A mutant abrogates almost completely the phosphorylation of the protein (Fig. 3B). Correspondingly, in cellulo, S10 is the most prominent GABARAPL2 phosphorylation site. Cell extracts of HEK293T cells overexpressing TBK1 and GABARAPL2 WT or non phosphorylatable alanine mutants were analyzed on a PhosTag gel, where phosphorylated proteins are retained and are represented by an upwards shift. Mutation of GABARAPL2 S10A results in a loss of phosphorylated protein (Fig. S3B). Similarly, mutation of TBK1 K38A (kinase dead mutant) abolished GABARAPL2 phosphorylation (Fig. S3C).

To understand the relevance of this phosphorylation we transfected HEK293T cells with different constructs of GABARAPL2, performed pulldowns of the mutants and used proteomics to analyze the variation in their interactomes (Fig. 3C). GABARAPL2 WT interacts with known partners belonging to biological processes such as macroautophagy, selective autophagy and intracellular transport[21,22], as shown with Gene Ontology Enrichment analysis (Fig. S3D). Comparison of the three pulldowns highlights that the phospho-mimicking mutant S10D loses affinity to several proteins compared to WT and S10A, which share the majority of interactors (Fig. 3D). The interaction of GABARAPL2 S10D to IRGQ, ULK1, ATG13, ATG4B and WDFY3 is significantly decreased, together with a small reduction in binding to ATG7, and p62/SQSTM1 (Fig. 3E). The loss in affinity to IRGQ was expected due to the previously described interaction mode of GABARAPL2 N-terminal to IRGQ, and was confirmed by WB of different pulldowns (Fig. S3E, F). To validate the MS findings on the other autophagy-related proteins, we made use of reconstituted cell lines from GABARAPL2 KO cells where the HA-GABARAPL2 WT or mutant genes were reintroduced with lentiviral transductions (Fig. S3G). HA-GABARAPL2 WT interacts with IRGQ, ATG7, and ULK1 under both basal and starvation conditions, whereas the HA-GABARAPL2 S10D mutant shows a pronounced reduction in IRGQ binding, with modest effects on ATG7 and ULK1 interactions (Fig. 3F). This selective loss of IRGQ engagement is consistent with our interactome analysis (Fig. 3E) and with the reduced association of these factors observed upon IRGQ knockdown (Fig. 1G). Autophagy-related proteins such as ULK1 typically bind ATG8 family members through LIR motifs that interact with the LDS[23,24]. S10 in GABARAPL2 is located at a considerable distance from the LDS (Fig. S3H) and its phosphorylation selectively impairs IRGQ binding. Because IRGQ likely stabilizes or promotes the assembly of complexes with ATG7 and ULK1, the weakened IRGQ-GABARAPL2 interaction in the S10D mutant may indirectly account for the subtle changes observed with these autophagy-initiation factors. This supports a model in which IRGQ functions as an interaction hub coordinating GABARAPL2 with components of the autophagy-initiation machinery.

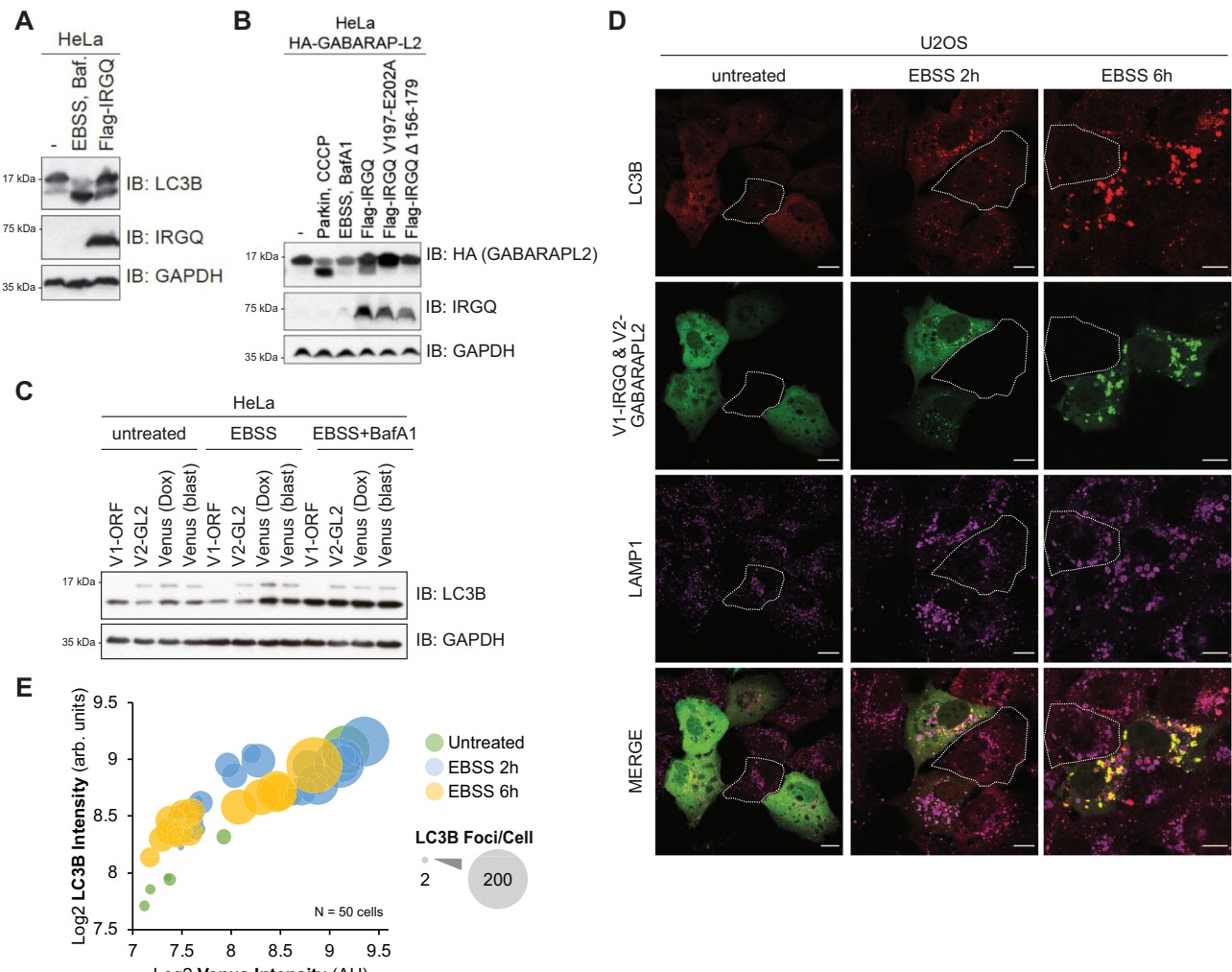

**Fig. 2 | The IRGQ-GABARAPL2 complex promotes hATG8 lipidation. A** SDS-PAGE and Western blot of HeLa cell lysates expressing transfected Flag-IRGQ or treated with EBSS, 200 nM Bafilomycin A1 (3 h); $n = 1$. **B** SDS-PAGE and Western blot of endogenously tagged HA-GABARAPL2 HeLa cell lysates with transfected Flag-IRGQ WT or mutants, treated with EBSS, 200 nM Bafilomycin A1 (3 h) or treated with CCCP 40 μM (3 h); $n = 1$. **C** SDS-PAGE and Western blot of HeLa cell lysates stably expressing V1-ORF, V2-GABARAPL2, V1-IRGQ or V1-IRGQ & V2-GABARAPL2 (Venus). Cells were either left untreated or treated with EBSS or EBSS and BafA1 (200 nM) for 3 h; $n = 1$. **D** Immunofluorescence of HeLa cells, stably expressing V1-IRGQ and V2-GABARAPL2. Autophagy was induced by the addition of EBSS for 2 h or 6 h. In addition, cells were left untreated. Fixed cells were probed with an endogenous LC3B or LAMP1 antibody. Scale bar: 20 μm. **E** Scatter plot of ImageJ quantification from (**D**). Single cells were annotated as ROIs and intensities and puncta for LC3B and Venus were measured. The boxed cells are non-transfected control cells for comparison to the transfected cells. RawIntDen values are plotted as Log2. Colors indicate the treatment and size of bubbles indicate the number of LC3B puncta per cell; $n = 50$ cells. Source data are provided as a Source data file.

To provide structural insights into how TBK1-dependent phosphorylation modulates this selective interaction, we performed molecular modeling of phosphorylated GABARAPL2 in the context of the IRGQ-ATG8 interface and additional ATG8-dependent assemblies. These models position S10 within the IRGQ-binding surface of GABARAPL2, such that the phosphate group (or the S10D phosphomimetic variant) perturbs the local electrostatic and steric environment at the IRGQ interface, thereby weakening IRGQ engagement. Importantly, our modeling showed that S10 of GABARAPL2 is spatially separated from the canonical LDS (LIR docking site), consistent with the experimental observation that LDS-mediated interactions of GABARAPL2 with other ATG8 interactors are largely preserved. Together, these data (Fig. S4A–H) point to a mechanism where TBK1 phosphorylation selectively destabilizes the IRGQ-GABARAPL2 complex while preserving broader ATG8 interactomes.

Consistent with this model, AlphaFold2-multimer predictions indicate that IRGQ can organize into higher-order assemblies with ATG8 proteins and the lipidation machinery. In predicted IRGQ-GABARAPL2-ATG7 complex models, IRGQ majorly engages with the canonical LDS of the GABARAPL2 interface (>70%). By contrast, fewer models display alternative configurations where ATG7 directly interacts with GABARAPL2 (<20%) (Fig. S5A). Extending this modeling to a four-component system, IRGQ-GABARAPL2-LC3B-ATG7 predominantly (>70%) place ATG7 on GABARAPL2 LDS while IRGQ engages with LC3B, with only a minor fraction showing swapped binding mode (ATG7 bound to LC3B and IRGQ bound to GABARAPL2) (Fig. S5B). Together, these models support our experimental data and indicate a scaffold mechanism where IRGQ, through its dual-LIR architecture, can simultaneously engage GABARAPL2 and LC3B, thereby promoting proximity to ATG7 and other ATG8-dependent factors without requiring direct IRGQ-ATG7 binding. Rather than directly abrogating LDS-mediated interactions, phosphorylation of GABARAPL2 at S10 by TBK1 selectively disrupts the IRGQ-GABARAPL2 interface of these scaffold complexes, indirectly destabilizing the assembly to reduce the overall recruitment efficiency of ATG8-associated machinery.

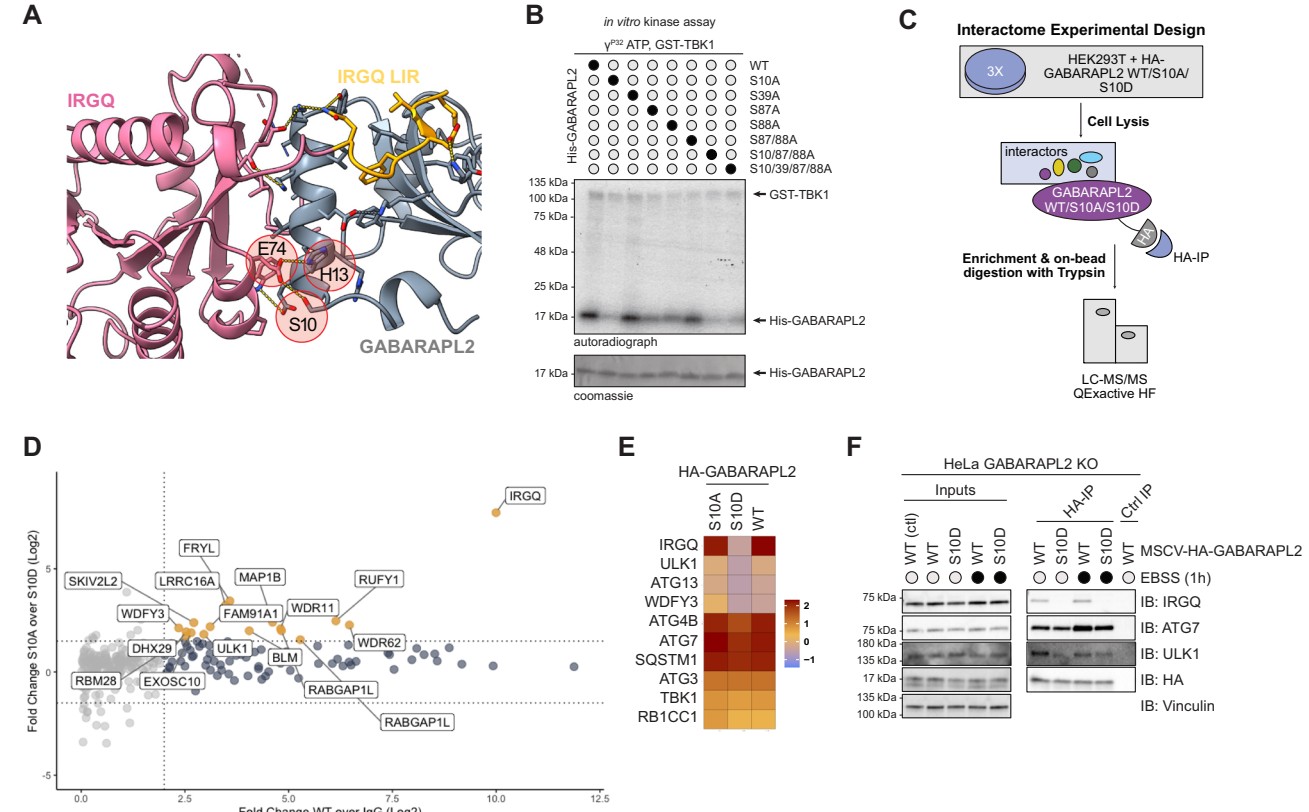

**Fig. 3 | IRGQ-GABARAPL2 complex stability is disrupted upon GABARAPL2 S10D mutation. A** Structure of IRGQ N-terminal domain (pink) and IRGQ LIR1 185-190 (yellow) in complex with GABARAPL2 (gray). GABARAPL2 α-helix 2 makes contacts within a pocket formed by IRGQ's switch I (partially disordered), switch II and the linker between LIR1 and N-terminal G-domain. Highlighted are key residues for this additional binding interface (E74 from IRGQ, H13 and S10 from GABAR-APL2). **B** Coomassie stain and autoradiography of SDS-PAGE after an in vitro kinase assay with GST-TBK1, His-GABARAPL2 WT and mutant proteins as substrates; $n = 1$. **C** Experimental set-up for label free interactome studies of the GABARAPL2 S10 mutants. HEK293T cells were transfected with HA-GABARAPL2 WT/S10A or S10D mutants, lysates used for HA-IPs and processed for mass spectrometry. Data was

analyzed with MaxQuant and Perseus; $n = 3$. **D** Scatter plot representing the Student's two-sided t-test difference from HA-GABARAPL2 over empty control IPs and the Student's two-sided t-test difference from HA-GABARAPL2 S10A over HA-GABARAPL2 S10D IPs. Significant and common interaction partners are marked in blue and interaction partners showing preference of binding towards S10A mutant are marked in yellow (Fold Change S10A over S10D ≥ 2). **E** Heatmap of selected autophagy-related proteins. The color scale represents the z-score of the average LFQ Intensity. **F** SDS-PAGE and Western blot of HA-IPs from GABARAPL2 KO cells reconstituted with HA-GABARAPL2 WT of S10D mutant (with or without EBSS treatment for 1 h); $n = 1$. Source data are provided as a Source data file.

## TBK1 phosphorylation of GABARAPL2 S10 restricts the IRGQ-GABARAPL2 autophagy hub

TBK1 has been demonstrated to play a critical role in selective autophagy via the regulation of a variety of autophagy proteins[25,26]. In order to test if TBK1 can phosphorylate GABARAPL2 on S10 in cells, we raised an antibody that specifically detects GABARAPL2 pS10 (Fig. S6A). TBK1 kinase activity can be stimulated under conditions that elicit selective autophagy (such as mitophagy or xenophagy). Induction of mitophagy as well as xenophagy and IFNγ treatment strongly induced endogenous GABARAPL2 S10 phosphorylation, especially of the lipidated form (Figs. 4A and S6B). However, the induction of starvation-induced bulk autophagy, which does not activate TBK1, did not lead to a phosphorylation of GABARAPL2 (Figs. 4A and S6B).

The appearance of this modification was dependent on the presence of TBK1 and specifically its kinase activity, as knockdown or chemical inhibition with MRT67307 or BX795 blocked phosphorylation of S10 GABARAPL2 (Fig. S6C, D). Intriguingly, TBK1 preferentially phosphorylated PE-conjugated His-GABARAPL2 as seen in an in vitro kinase assay (Fig. S6E). To more closely delineate the kinetics of this phosphorylation, we induced mitophagy and measured GABARAPL2 pS10 at various time points. GABARAPL2 pS10 peaks at 1 h after treatment and coincides with maximal TBK1 activation (Figs. 4B and S6F). Similarly, infection with *Salmonella* to trigger TBK1 activation

results in phosphorylation of S10 GABARAPL2, with similar kinetics as mitophagy induction (Fig. S6G). Notably, GABARAPL2 pS10 can also be detected in human samples from HBV-infection derived tumors (Fig. S6H)[27].

Since phospho-mimicking mutations of GABARAPL2 S10 restricted its interaction with IRGQ, we confirmed that TBK1 activation in cells causing phosphorylation of S10 on GABARAPL2 had the same effect on its binding to endogenous IRGQ. Indeed, TBK1 stimulation upon xenophagy induction causes a significant reduction in IRGQ-GABARAPL2 endogenous foci measured with proximity ligation assays (PLA) (Figs. 4C and S6I, J). Furthermore, co-immunoprecipitation of endogenous GABARAPL2 with IRGQ is substantially restricted upon TBK1 stimulation as seen by Western Blot (Figs. 4D and S6K). To further validate the effect of TBK1 on the regulation of the IRGQ-GABARAPL2 hub, HEK293T cells were co-transfected with Flag-IRGQ and HA-GABARAPL2 WT, S10D or S10A mutant constructs and subjected to a time course treatment of CCCP to trigger TBK1 activation. In basal conditions, HA-GABARAPL2 WT interacts strongly with Flag-IRGQ but the complex is disrupted upon CCCP treatment, with the lowest interaction signal coinciding with the highest peak of TBK1-mediated S10 phosphorylation of GABARAPL2 at 1 h after treatment (Fig. 4B, E). In contrast, HA-GABARAPL2 S10A interaction with Flag-IRGQ is unaffected upon TBK1 activation,

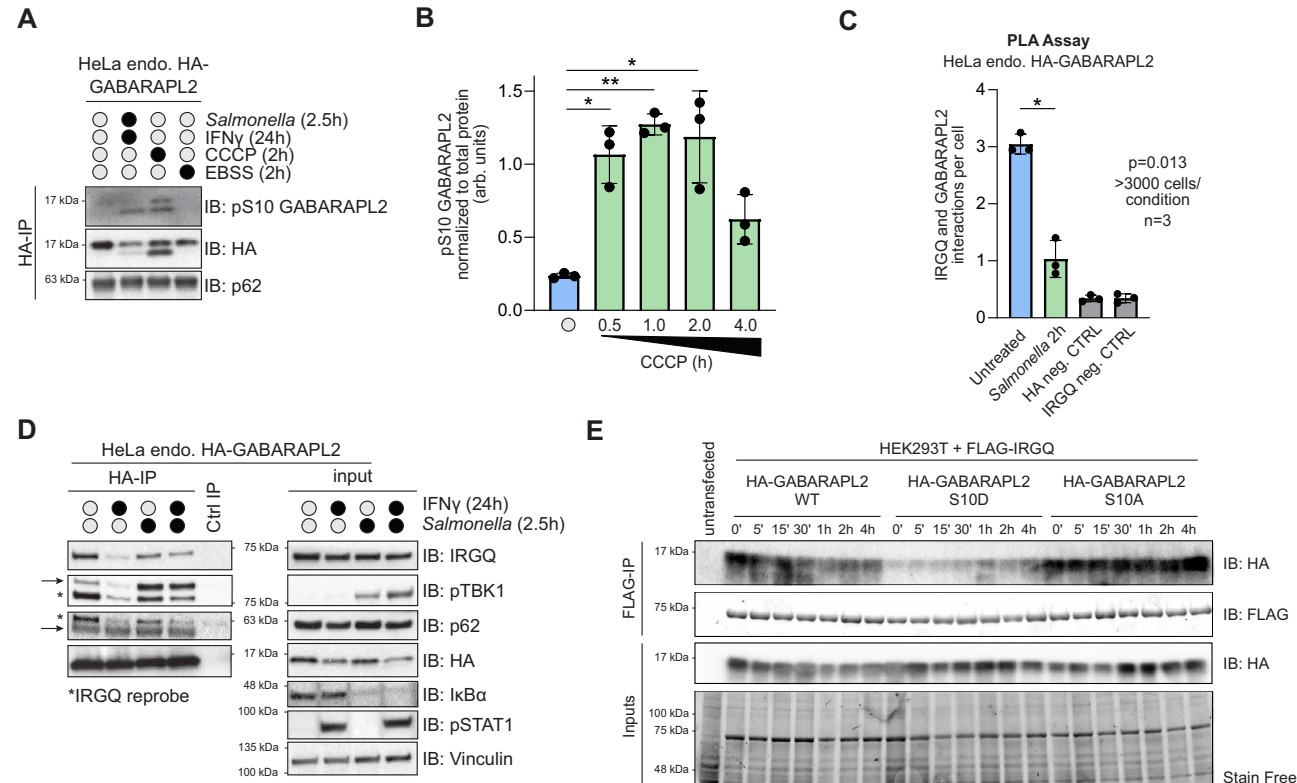

**Fig. 4 | TBK1 phosphorylation of GABARAPL2 S10 restricts the IRGQ-GABARAPL2 autophagy complex. A** SDS-PAGE and Western blot of HA-IPs from endogenously tagged HA-GABARAPL2 WT HeLa cells after treatment with CCCP (2 h), EBSS (2 h) and IFNγ (24 h) plus *Salmonella* infection (2.5 h). **B** ImageJ quantification from Figure S6F of pS10 GABARAPL2, normalized to total GABARAPL2 protein. Data are presented as the mean with error bars indicating the s.d. Statistical significance of differences between experimental groups was assessed with two-sided Student's t-test. Differences with $p < 0.05$ are annotated as * and $p < 0.01$ are annotated as **; $p$ value for 0.5 h = 0.02115609, $p$ value for 1 h = 0.00198796, $p$ value for 2 h = 0.03360764, $p$ value for 4 h = 0.05712305; $n = 3$ (biological replicates). **C** Yokogawa CQ1 quantification of average Duolink PLA signal from endogenous GABARAPL2 and endogenous IRGQ. HeLa endogenous HA-GABARAPL2 cells were

infected with *Salmonella* for 2 h and fixed cells were probed with the Duolink in situ PLA assay. HA and IRGQ only antibodies were used as negative controls to determine the background. Data are presented as the mean with error bars indicating the s.d. Statistical significance of differences between experimental groups was assessed with two-sided Student's t-test. Differences with $p < 0.05$ are annotated as * ($p$ value = 0.01380386); $n = 3$ (biological replicates); >3000 cells/condition. **D** SDS-PAGE and western blot of HeLa endogenous HA-GABARAPL2 cells that were treated with 10 ng/ml IFNγ (24 h) and infected with WT *Salmonella* (2.5 h). Cell lysates were subjected to HA-IPs or IgG control IP; $n = 1$. **E** SDS-PAGE and western blot of HEK293T exogenously expressing Flag-IRGQ and HA-GABARAPL2 WT or S10 mutants. Cells were treated with 40 μM CCCP at specified timepoints. Cell lysates were subjected to FLAG-IP; $n = 1$. Source data are provided as a Source data file.

showing a slight increase at the 4 h timepoint. As expected, HA-GABARAPL2 S10D mutant does not interact with Flag-IRGQ even in basal conditions (Fig. 4E). Hence, GABARAPL2 S10 is specifically modified by TBK1, once it is activated and causes IRGQ-GABARAPL2 complex disruption.

### TBK1 is a negative regulator of IRGQ-mediated autophagy
The IRGQ-GABARAPL2 hub interacts with the autophagy initiation machinery and facilitates the lipidation of LC3B. Upon TBK1 activation, GABARAPL2 is phosphorylated on S10 causing the dissociation from IRGQ and autophagy initiation proteins (Fig. 5A). Therefore, we tested if this regulatory modification would also impact LC3B lipidation or flux. To assess whether phosphorylation of GABARAPL2 at S10 impacts autophagy progression, we performed a high-throughput microscopy time-course assay coupled to automated image analysis in HA-GABARAPL2 WT, S10A, and S10D reconstituted cells. We quantified LC3-to-lysosome flux in cells expressing GABARAPL2 WT, S10A, or S10D (Fig. 5B). Each data point represents the normalized mean LC3-to-lysosome ratio per well and per biological replicate, normalized within each repeat to the untreated WT mean. Conditions included starvation (EBSS, 1–4 h), mitochondrial depolarization (40 μM CCCP, 1–4 h), and lysosomal inhibition (200 nM BAF for 4 h, ±EBSS or CCCP). Across these conditions, LC3-to-lysosome flux was not significantly altered by changes in GABARAPL2 S10 phosphorylation, indicating

that S10D does not measurably impair inducible bulk autophagy flux. Together, these data support the conclusion that the primary impact of the S10D mutation is on IRGQ engagement and the basal organization of ATG8-positive structures, rather than on global autophagy induction or lysosomal delivery of LC3 under stress.

In contrast to LC3B flux, GABARAPL2 autophagic flux itself was altered by the phospho-mimicking mutant. To monitor GABARAPL2 trafficking directly, we generated U2OS reporter cell lines expressing mCherry-GFP-tagged GABARAPL2 WT, S10A, or S10D and quantified flux using automated high-throughput imaging (Fig. S7A, B) and FACS (Fig. S7C, D). Upon EBSS starvation, WT and S10A cells showed an increase in GABARAPL2 flux, and this response was blocked by BafA1, consistent with lysosome-dependent turnover. By contrast, S10D cells exhibited lower basal GABARAPL2 flux, and starvation failed to further enhance flux (Fig. S7A). Across both readouts, S10A closely phenocopied WT, whereas S10D displayed a selective flux defect consistent with a constitutive phospho-mimetic state. These data support a model in which TBK1-mediated phosphorylation of GABARAPL2 at S10 functions as a conditional negative regulatory ("stop") signal, terminating IRGQ-GABARAPL2 engagement and limiting lysosomal trafficking once the relevant step of the pathway has been completed.

To validate that the S10D mutation disrupts delivery of the GABARAPL2-IRGQ complex to lysosomes, we performed immunofluorescence analyses. Under starvation-induced autophagy, GFP-

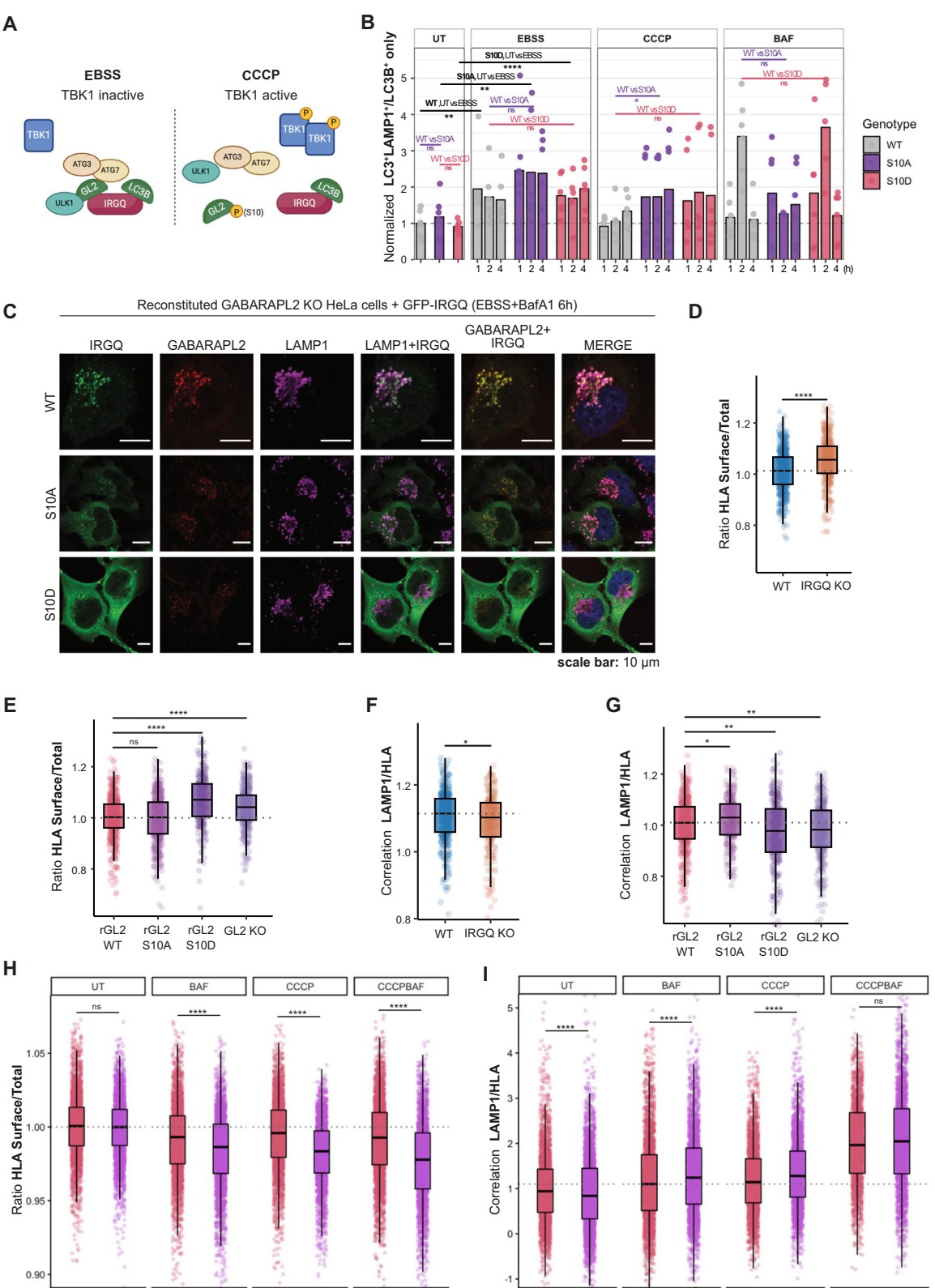

IRGQ formed punctate structures that colocalized with GABARAPL2 and LAMP1 in reconstituted GABARAPL2 WT cells (Fig. 5C). Similarly, GFP-IRGQ puncta colocalizing with LC3B and LAMP1 were observed in GABARAPL2 S10A cells (Fig. 5C). In contrast, in S10D cells GFP-IRGQ remained diffusely cytosolic and failed to form puncta (Fig. 5C), underscoring that a tightly regulated S10 phosphorylation state is required for IRGQ-dependent autophagy. In line with this model, WT

GABARAPL2 is expected to be largely unphosphorylated under basal conditions and phosphorylation to be transient and context-dependent, providing a mechanistic explanation for why the non-phosphorylatable S10A mutant often resembles WT.

Recently, we described IRGQ as an autophagy receptor that functions in the quality control of 'non-conformational' HLA molecules[16]. Naturally, we tested if this function of IRGQ was also

**Fig. 5 | TBK1 is a negative regulator of IRGQ-mediated autophagy. A** Model of IRGQ autophagy hub in basal conditions or upon TBK1 activation. Created in BioRender. Gestal mato, U. (2026) https://BioRender.com/6sufx0v. **B** LC3-to-lysosome flux under stress is not changed when GABARAPL2 S10 phosphorylation is altered. Quantification of LC3-to-lysosome flux in GABARAPL2 WT, S10A, and S10D cells. Biological replicates: $n = 3$. Technical replicates: 2 wells per condition per experiment. Each point represents the mean per well, normalized to the untreated (UT) WT mean within the same experiment. Conditions represent starvation (EBSS, 1–4 h), mitochondrial depolarization (40 μM CCCP, 1–4 h), or lysosomal inhibition (200 nM bafilomycin A1 (BAF), 4 h; ±EBSS or CCCP). Statistics: two-sided one-sample t-tests comparing the indicated grouped conditions to UT ( = 1). **** $p < 0.0001$; *** $p < 0.001$; ** $p < 0.01$; * $p < 0.05$; n.s., not significant. Exact $p$ values are provided in the Source data file. **C** Immunofluorescence of HeLa GABARAPL2 KO cells reconstituted with WT, S10A or S10D HA-GABARAPL2 and overexpressed GFP-IRGQ. Autophagy was induced by the addition of EBSS for 6 h and Bafilomycin A1 (200 nM). Fixed cells were probed with endogenous LAMP1 and HA antibodies. Scale bar: 10 μm. **D, E** Normalized LAMP1/HLA correlation. Automated single-cell quantification using CellProfiler. Biological replicates: $n = 2$. Technical replicates: 3 wells per condition per experiment. Each dot represents one cell from an individual well. Data were normalized to the UT mean within each biological replicate. Box plots: center line = median; box bounds = 25th–75th percentiles (IQR); whiskers extend to the most extreme values within 1.5×IQR; outliers beyond 1.5×IQR are not displayed. Statistics: two-sided Wilcoxon rank-sum test for the indicated comparisons; multiple-comparison correction using Benjamini–Hochberg FDR. **** $p < 0.0001$; *** $p < 0.001$; ** $p < 0.01$; * $p < 0.05$; n.s. not significant. Exact $p$ values are provided in the Source data file. **F, G** Normalized HLA surface/total intensity. Automated single-cell quantification using CellProfiler after 4 h treatment as indicated (EBSS and/or 200 nM BAF). Biological replicates: $n = 2$. Technical replicates: 3 wells per condition per experiment. Each dot represents one cell from an individual well. Data were normalized to the UT mean within each biological replicate. Box plots: center line = median; box bounds = 25th–75th percentiles (IQR); whiskers extend to the most extreme values within 1.5×IQR; outliers beyond 1.5×IQR are not displayed. Statistics: two-sided Wilcoxon rank-sum test; multiple-comparison correction using Benjamini–Hochberg FDR. **** $p < 0.0001$; *** $p < 0.001$; ** $p < 0.01$; * $p < 0.05$; n.s. not significant. Exact $p$ values are provided in the Source data file. **H, I** Quantification of HLA-to-lysosome flux and HLA presentation at the cell surface in HeLa WT cells. Biological replicates: $n = 2$. Technical replicates: 3 wells per condition per experiment. Each point represents the mean per well, normalized to the UT mean within the same experiment. Conditions include mitochondrial depolarization (40 μM CCCP, 4 h) or lysosomal inhibition (200 nM BAF, 4 h) ± TBK1 inhibition (MRT67307, 5 μM). Statistics: two-sided Wilcoxon rank-sum test; multiple-comparison correction using Benjamini–Hochberg FDR. **** $p < 0.0001$; *** $p < 0.001$; ** $p < 0.01$; * $p < 0.05$; n.s. not significant. Exact $p$ values are provided in the Source data file. Source data are provided as a Source data file.

negatively regulated by the TBK1-mediated phosphorylation of GABARAPL2, since flux of IRGQ to the lysosome is impaired upon GABARAPL2 depletion[16]. Immunofluorescence analysis and quantification with CellProfiler confirmed that HLA (IRGQ's cargo) was accumulated at the cell surface of S10D mutants, mimicking the phenotype of IRGQ KO cells (Figs. 5D, E and S7D). This was seemingly a cause of impaired lysosomal delivery of HLA molecules observed in these cells (Figs. 5F, G and S7D). In addition, we confirmed that pharmacological inhibition of TBK1 using MRT67307 further enhances lysosomal delivery of HLA molecules (Fig. 5H, I). Specifically, we quantified HLA-to-lysosome flux under stress conditions in the presence of MRT67307 and observed increased HLA accumulation in lysosomal compartments, consistent with our findings in Fig. 5D–G. Together, these results corroborate that TBK1 activity restrains IRGQ-dependent trafficking of HLA to lysosomes, and that TBK1 inhibition promotes HLA turnover via the lysosomal pathway under stress conditions.

To determine whether GABARAPL2 S10 phosphorylation broadly affects autophagy or selectively impacts IRGQ-dependent pathways, we assessed p62 turnover as a readout of bulk and parallel selective autophagy fluxes. We quantified p62-to-lysosome flux in cells expressing GABARAPL2 WT, S10A, or S10D under starvation (EBSS), mitochondrial depolarization (CCCP), and lysosomal inhibition (BAF) conditions (Fig. S7F). p62 degradation was comparable between WT, S10A, and S10D cells, indicating that p62-dependent autophagy flux is independent of GABARAPL2 S10 phosphorylation. These data demonstrate that S10 phosphorylation does not impair bulk autophagy (consistent with no impact on LC3B flux, Fig. 5B), but instead selectively affects IRGQ-dependent cargo turnover, exemplified by MHC-I degradation (Fig. 5H, I).

In short, disruption of the IRGQ-GABARAPL2 autophagic hub by TBK1 stimulation as well as phospho-mimicking mutations of GABARAPL2 S10 affected the flux of GABARAPL2 and IRGQ's cargo molecule HLA, but not bulk autophagy, revealing TBK1 as a negative regulator of IRGQ's autophagic axis.

## Discussion

Autophagy is a tightly regulated catabolic pathway critical for cellular homeostasis, with selective autophagy receptors serving as key coordinators between cargo recognition and the core autophagy machinery[28]. Our study identifies the IRGQ-GABARAPL2 complex as a central organizer of autophagy initiation, extending the functional repertoire of IRGQ beyond cargo recognition to active orchestration of GABARAPL2 lipidation and flux. While IRGQ was previously characterized as a receptor for misfolded MHC class I molecules[16], we now demonstrate that it additionally functions as a scaffold to promote autophagy initiation and is subject to regulation by TBK1.

We show that IRGQ binds to both GABARAPL2 and LC3B via two conserved LIR motifs, supporting its role as a dual hATG8 interactor. This dual engagement enables IRGQ to promote the recruitment and lipidation of GABARAPL2 and LC3B, thereby driving autophagosome formation. The importance of this interaction is underscored by the finding that IRGQ localizes to lysosomes in an autophagy-dependent manner, suggesting efficient flux under basal conditions.

Unexpectedly, we uncover that TBK1, which is widely recognized as a positive regulator of autophagy[29,30], can act as a negative regulator in this context. TBK1 activation leads to phosphorylation of GABARAPL2 at S10, disrupting its interaction with IRGQ and thereby disassembling the complex. This, in turn, impairs the flux of GABARAPL2 and IRGQ's cargo, misfolded MHC-I, to lysosomes, without affecting bulk autophagy. These findings reveal a previously unrecognized mechanism by which TBK1 fine-tunes autophagic responses through spatial and temporal control of hub assembly.

Importantly, this regulatory axis challenges the prevailing view of TBK1 as a uniformly pro-autophagic kinase. While TBK1-mediated phosphorylation of autophagy receptors is often associated with enhanced flux[31], our data show that it can also suppress selective autophagy by dismantling key protein-protein interactions. Thus, the functional outcome of TBK1 activity appears to be highly context-dependent, varying with the identity of the autophagy receptor and cargo.

In this context, we propose that TBK1-mediated phosphorylation of GABARAPL2 at S10 acts as a negative regulatory termination signal. Specifically, phosphorylation at S10 disrupts IRGQ–GABARAPL2 engagement and thereby limits further trafficking of GABARAPL2-containing complexes to lysosomes once the relevant step of the pathway has been executed. Such a mechanism implies that S10 phosphorylation is transient and conditional, which provides a straightforward explanation for why the non-phosphorylatable S10A mutant often phenocopies WT: under basal conditions, GABARAPL2 is expected to be largely unphosphorylated, and thus WT and S10A behave similarly. By contrast, S10D mimics constitutive phosphorylation, enforcing a persistent "off" state that blocks IRGQ binding and downstream lysosomal targeting, thereby revealing the importance of regulated disengagement rather than continuous interaction.

AMPK is a well-established energy sensor and a key regulator of autophagy, traditionally recognized for its role in promoting autophagy under conditions of energy stress. However, recent findings underscore a more nuanced function of AMPK, revealing its dual role as both a promoter and inhibitor of autophagy depending on the cellular context and regulatory interactions[32]. This duality parallels our findings, where we now demonstrate that TBK1, similarly known for its autophagy-promoting activities, can also function as a negative regulator by inhibiting the binding of GABARAPL2 to IRGQ. These insights highlight the context-dependent and multifaceted roles of kinases in autophagy regulation.

Our work also provides mechanistic insights into the control of MHC class I quality via autophagy. By demonstrating that IRGQ facilitates not only cargo recognition but also initiation of autophagy, we suggest that certain receptors act as integrators of cargo selection and machinery assembly. This positions IRGQ at the nexus of immune regulation and proteostasis, with implications for understanding immune evasion, antigen presentation, and inflammatory signaling.

In summary, we propose a model in which the IRGQ-GABARAPL2 complex functions as a scaffolding complex that nucleates the autophagy initiation complex through its dual hATG8 interactions and the recruitment of the autophagy-initiation machinery, and whose activity is negatively regulated by TBK1-mediated phosphorylation of GABARAPL2. These findings expand our understanding of selective autophagy regulation and reveal additional layers of control by kinase signaling networks.

## Methods

### Expression constructs
Expression constructs of indicated proteins were cloned into indicated vectors using PCR or the gateway system. Site-directed mutagenesis was performed by PCR to introduce desired amino acid substitutions. All expression constructs were sequenced by Seqlab.

### Cell culture
HEK293T, U2OS, and HeLa cells were cultured in Dulbecco's modified Eagle's medium (DMEM; Gibco) supplemented with 10% fetal bovine serum (FBS), 1% penicillin/streptomycin and maintained at 37 °C in a humidified atmosphere with 5% $CO_2$. For *Salmonella* infections penicillin/streptomycin was omitted from the media. HeLa WT endogenous HA-GABARAPL2 cells were generated using CRISPR/Cas9 technology and kindly provided by the Behrends laboratory (LMU München, Germany). The guide RNAs used for the CRISPR/Cas 9 KO cells were designed using the GPP Web Portal of the Broad Institute (http://www.broadinstitute.org/rnai/public/analysis-tools/sgrna-design). These are: IRGQ #1: TTTGTGCTACCGGCGAACTG, IRGQ #2: GAATGCACTCAGT AAGGGAA, IRGQ #3: CGTGAGGCCTTTGAGACCGG, GABARAPL2 #1: GTCGAGCGAAATATCCCGACA, GABARAPL2 #2: GTCCCACAGAACAC AGATGCG, GABARAPL2 #3: GGTTCCATCTGATATCACTG, and were cloned into Lentiviral vectors containing CAS9: pLenti-Puro or pLenti-Neo. This was done as described previously[33]: Lentiviral vectors containing Cas9 (pLenti-Puro, pLenti-Puro-EGFP, or pLenti-Neo) were digested with the restriction enzyme BsmBI and gel-purified. For ligation, 100 ng of the linearized vector was combined with 4 µl of annealed oligonucleotides (diluted 1:200) and incubated with T4 DNA ligase for 1 h at room temperature. The ligation products were subsequently transformed into XL1-Blue competent cells. Lentiviral particles were produced in HEK293T cells. Cells were seeded in six-well plates and cultured in 2 ml DMEM supplemented with 10% FBS until approximately 90% confluence. HEK293T cells were then cotransfected with three lentiviral plasmids encoding Cas9 and gene-specific sgRNAs (1.1 µg DNA per plasmid), together with the packaging plasmids pPAX2 (2.2 µg DNA) and pMD2.G (1 µg DNA). Viral supernatants were collected 24 h post-transfection, and the culture medium was replaced with 2 ml of fresh DMEM. A second collection was performed after an additional 24 h. The pooled viral supernatants (4 ml total) were centrifuged to remove cellular debris and stored at −80 °C. To generate knockout cell lines, target cells were infected with 1 ml of lentiviral supernatant containing three sgRNA constructs targeting the gene of interest. The remaining viral supernatant was stored at −80 °C. Forty-eight hours after infection, cells were selected in media containing 2 µg/ml of Puromycin or 1 mg/ml Neomycin. To create the reconstituted cell lines, HeLa GABARAPL2 KO cells (pLenti-Neo) were used and lentiviral transductions of HA-GABARAPL2 WT or mutants were performed, with a selection of 2 µg/ml of Puromycin or 1 mg/ml Neomycin.

CCCP was resuspended in DMSO and cells were treated with 40 µM for specific timepoints (30 min–4 h). Human IFNγ (AF-300-02; Peprotech) was added to cells for 24 h at a final concentration of 10 ng/ml. Nutrient starvation was achieved by replacing DMEM with EBSS (Gibco) for 30 min–4 h. BafilomycinA1 was resuspended in DMSO and cells were treated with 200 nM for specific timepoints. MRT67307 (a specific TBK1 inhibitor) was used at 5 µM for 4 h.

Plasmid transfections were performed with 3 µl GeneJuice (Merck Millipore), 0.5 µg plasmid DNA in 200 µl Opti-MEM (Life Technologies). After incubation for 15 min, the solution was added to the cells, which were lysed in lysis buffer or fixed with 4% paraformaldehyde 48 h later.

SiRNA transfections were performed with 3 µl RNAiMax (Invitrogen), 20 nM siRNA (IRGQ #1: TTTGTGCTACCGGCGAACTG, IRGQ #2: GAATGCACTCAG-TAAGGGAA, IRGQ #3: CGTGAGGCCTTTGAGA CCGG; TBK1 #1: 5′-GACAGAAGUUGUGAUCACATT-3′) all purchased from Sigma) in 150 µl Opti-MEM (Life Technologies). After incubation for 30 min, the solution was added to the cells cultured in a 6-well dish, which were lysed in lysis buffer or fixed with 4% paraformaldehyde 72 h post transfection.

### Salmonella infections
Salmonella SL1344 (WT) were streaked out on LB plates and single colonies were picked to inoculate 2 ml of LB media (containing appropriate antibiotics and 0.3 M NaCl) to grow at 37 °C for 16 h. The overnight culture was then diluted 1:33 in LB media (containing appropriate antibiotics and 0.3 M NaCl). After 2.5 h, the OD600 of *Salmonella* was determined and the cells were infected for 30 min with a MOI of 150 (considering that OD600 = 1 has -1.3 × 10$^9$ bacteria/ml). The infection media was then exchanged to DMEM (+10% FBS) with 50 µg/ml Gentamycin and cells were lysed or fixed after indicated time points.

### Immunofluorescence for confocal microscopy imaging
HeLa or U2OS cells were seeded onto glass coverslips in 12-well culture dishes and treated accordingly. Cells were washed in phosphate-buffered saline (PBS) before fixation with 4% paraformaldehyde for 15 min at room temperature. The coverslips were washed a further three times before permeabilization of the cells with 0.5% Triton X-100 in PBS for 10 min at room temperature. Cells were rinsed with PBS before being incubated for 1 h in 1% bovine serum albumin (BSA) in PBS for 1 h. Primary antibody incubation was done for 1 h in a humidified chamber with 1% BSA in PBS. After thorough washes in PBS, cells were incubated with secondary antibodies, 1% BSA in PBS for 1 h in the dark. Cells were washed three more times in PBS and once with deionized water before being mounted onto glass slides using ProLong Gold mounting reagent (Life Technologies), which contained the nuclear stain 4′,6-diamidino-2-phenylindole (DAPI). Slides were imaged using a Leica microscope Confocal SP 80 fitted with a 60x oil-immersion lens.

### Immunofluorescence for Yokogawa CQ1 microscopy imaging
HeLa or T-REx mCherry-GFP GABARAPL2 WT, S10A and S10D mutant U2OS cells were seeded onto black, clear flat bottom 24- or 96-well plates (2000 cells/well for 96-well plates and 18,000 cells/well for

24-well plates). When indicated, cells were treated. Cells were washed in PBS before fixation with 4% paraformaldehyde for 15 min at room temperature. Cells were rinsed with PBS before being incubated for 1 h in permeabilization and primary antibody solution (0.1% Saponin (47036; Sigma), 5 mM MgCl₂, 5% BSA in PBS). After washes in PBS, cells were incubated with Hoechst 33342 (R37605; Thermo Fisher), Alexa Fluor 647 Phalloidin (#8940; Cell Signaling Technology) and Alexa Fluor secondary antibodies in antibody solution (0.1% Saponin (47036; Sigma), 5 mM MgCl₂, 5% BSA in PBS) for 1 h in the dark. Plates were imaged using a Yokogawa CQ1 microscope.

## Cell lysis

For lysis, cells were washed with PBS and scraped on ice in IP lysis buffer (50 mM Hepes, pH 7.5, 150 mM NaCl, 1 mM EDTA, 1 mM EGTA, 1% Triton X-100, 25 mM NaF, 5% glycerol, 10 µM ZnCl2) or total cell lysis buffer (50 mM Tris HCl, pH 7.5, 1 mM EDTA, 1% SDS, 25 mM NaF, 1 µl/ml Benzonase (71205-25KUN; Millipore)), both supplemented with complete protease inhibitors (cOmplete, EDTA-free; Roche Diagnostics) and phosphatase inhibitors (P5726, P0044; Sigma). Extracts were cleared by centrifugation at $21000 \times g$ for 15 min at 4 °C.

## Immunoprecipitation of overexpressed proteins

Cleared cell extracts were mixed with HA-agarose beads (A2095; Sigma), Flag-M2 agarose beads (A2220; Sigma), RFP-Trap_A beads (rta-10; ChromoTek) or GFP-Trap_A beads (gta-10; ChromoTek) 16 h at 4 °C on a rotating platform. The beads were washed four times in IP lysis buffer. Immunoprecipitated and input samples were reduced in SDS sample buffer (50 mM Tris HCl, pH 6.8, 10% glycerol, 2% SDS, 0.02% bromophenol blue, 5% β-mercaptoethanol) and heated at 95 °C for 5 min[34].

## Immunoprecipitation of endogenous HA-GABARAPL2

HeLa cells stably expressing HA-tagged GABARAPL2 were seeded in 15-cm dishes and cultured to ~80% confluency prior to lysis. Cells were treated as indicated (e.g., siRNA knockdown, CCCP, or *Salmonella* infection). Cells were washed once with 10 mL PBS and lysed in 150 µL Total Cell Lysis (TCL) buffer (1% SDS, 50 mM Tris-HCl pH 7.5, 1 mM EDTA, 25 mM NaF, supplemented with protease inhibitor cocktails 2 and 3). Lysates were immediately transferred into 2 mL Eppendorf tubes, diluted with 1.35 mL of Normal Lysis (NL) buffer (150 mM NaCl, 50 mM HEPES pH 7.5, 1 mM EDTA, 1 mM EGTA, 10% (v/v) glycerol, 1% (v/v) Triton X-100, 10 µM ZnCl₂, 25 mM NaF, supplemented with protease inhibitor cocktails 2 and 3), and placed on ice. To degrade nucleic acids and reduce viscosity, 1 µL of Benzonase (Sigma–Aldrich) was added to each lysate, followed by incubation on ice for 20 min. Lysates were clarified by centrifugation at maximum speed for 10 minutes at 4 °C.

Input samples were prepared by transferring 50 µL of the cleared supernatant into a fresh tube, adding 10 µL of 4× SDS sample buffer, heating for 10 min at 95 °C, and storing at −20 °C.

HA immunoprecipitations were set up by incubating the remaining lysates with 20 µL of pre-equilibrated anti-HA magnetic beads (e.g., Thermo Fisher) overnight at 4 °C with gentle rotation in 1.5 mL tubes. The following day, beads were washed four times with chilled NL buffer. After the final wash, beads were dried using a 27 G ¾" needle to minimize residual buffer volume. Proteins were eluted by adding 20 µL of 2× SDS sample buffer directly to the beads, followed by heating at 95 °C for 10 min.

## Protein binding assays

GST or GST-IRGQ were immobilized on glutathione-Sepharose beads (GE Healthcare) and combined with purified His-GABARAPL2 in protein binding buffer (150 mM NaCl, 50 mM, Tris, pH 7.5, 0.1% Nonidet P-40, supplemented with 5 mM DTT and 0.25 mg/mL BSA). The proteins were incubated on a rotating platform at 4 °C for 16 h. After five washes with buffer, proteins were diluted with SDS sample buffer

(62.5 mM Tris-HCl pH 6.8, 10% (v/v) glycerol, 2% (w/v) SDS, 0.02% (w/v) bromophenol blue, 5% (v/v) β-mercaptoethanol), resolved by SDS-PAGE and analyzed by immunoblotting with the indicated antibodies.

## Kinase assays

GABARAPL2 WT and mutant proteins were incubated in 20 µl phosphorylation buffer (50 mM Tris HCl, pH 7.5, 10 mM MgCl₂, 0.1 mM EGTA, 20 mM ß-glycerophosphate, 1 mM DTT, 0.1 mM Na₃VO₄, $\gamma^{P32}$ ATP (500 cpm/pmol; SRP-201; Hartmann Analytic)) with 50 ng of recombinant GST-TBK1 for 15 minutes at 30 °C. The kinase assay was stopped by adding SDS sample buffer containing 1% β-mercaptoethanol and heating at 95 °C for 5 min. The samples were resolved by SDS-PAGE, and the gels were stained with InstantBlue (expedeon) and dried. The radioactivity was analyzed by autoradiography[35].

## Western blotting

For immunoblotting, proteins were resolved by SDS-PAGE and transferred to PVDF membranes. Blocking and primary antibody incubations were carried out in 5% BSA in TBS-T (150 mM NaCl, 20 mM Tris, pH 8.0, 0.1% Tween-20), secondary antibody incubations were carried out in 5% low-fat milk in TBS-T and washings in TBS-T. Blots were developed using Western Blotting Luminol Reagent (sc-2048; Santa Cruz). Immunoblot bands were quantified using ImageJ software. All Western blots shown are representative.

## Antibodies

The following antibodies were used in this study: anti-HA-tag (11867423001; Roche), anti-FlagM2-tag (F3165; Sigma), anti-GFP-tag (Living Colors 632592; Clontech), anti-His-tag (11922416001; Roche), anti-vinculin (V4505; Sigma), anti-TBK1 (#3013; Cell Signaling Technology), anti-pTBK1 (pS172; #5483; Cell Signaling Technology), anti-IRGQ (HPA043254; Sigma), anti-GAPDH (#2118; Cell Signaling Technology), anti-GABARAPL2 (PM038; MBL), anti-LAMP1 (H4A3; DSHB), anti-p62 (M162-3; MBL), anti-IkBa (#9247; Cell Signaling Technology), anti-Histone H3 (ab1791, Abcam), anti-pSTAT1 (pY701; #7649; Cell Signaling Technology), anti-ATG3 (#3415; Cell Signaling Technology), anti-ATG7 (8558S, CST),anti-LC3B (PMO36; MBL), anti-ULK1 (8054S, CST), anti-MFN1 (14739S, CST), anti-pS10 GABARAPL2 (was generated by immunGlobe®, a chemically synthesized peptide (GABARAPL2 aa4-15) bearing a phosphate group at S10 (Ac-MFKEDH(pS)LEHRC-NH₂) was used for immunization). Primary antibodies used for Western blotting were diluted 1:1000 and for immunofluorescence studies 1:200. Secondary HRP conjugated antibodies goat anti-mouse (sc-2031; Santa Cruz), goat anti-rabbit (sc-2030; Santa Cruz) and goat anti-rat (sc-2006; Santa Cruz) were used for immunoblotting. Anti-rat Alexa Fluor 647 (A-21247; Life Technologies), anti-rat Cy3 (712166153; Jackson Lab), anti-mouse Alexa Fluor 405 (A-31553; Life Technologies), anti-mouse Cy3 (715-165-151; Dianova), anti-mouse Alexa 647 (A-31626; Life Technologies) were used for immunofluorescence studies.

## Mass spectrometry

Cells were treated, lysed and IP was performed as stated above, after which trypsin digestion and peptide desalting were performed. The Venus IP dataset (Figs. 1C–E and S1F) had a total of 12 samples of which 3 were control IPs and 9 were 3 different conditions with 3 technical replicates each. Similarly, HA-IP dataset (Figs. 3C–E and S3D) had a total of 12 samples of which 3 were control IPs and 9 were 3 different conditions with 3 technical replicates each. Digested peptides were acidified with trifluoroacetic acid (TFA) (Sigma Aldrich) to inhibit trypsin and to acidify peptides for SDB-RPS StageTip desalting. Acidified peptides were loaded onto the SDB-RPS StageTips and then washed with 0.1% (v/v) TFA. Peptides were eluted using a two-step elution with 0.1% (v/v) TFA, 80% (v/v) ACN and then dried using a speed-vacuum concentrator (30–45 min at 45–60 °C). Dried peptides were stored at −20 °C.

Samples were analyzed on a Q Exactive HF coupled to an easy nLC 1200 (ThermoFisher Scientific) using a 35 cm long, 75 μm ID fused-silica column packed in house with 1.9 μm C18 particles (Reprosil pur, Dr. Maisch), and kept at 50 °C using an integrated column oven (Sonation). Peptides were eluted by a non-linear gradient from 4 to 28% acetonitrile over 45 min and directly sprayed into the mass-spectrometer equipped with a nanoFlex ion source (ThermoFisher Scientific). Full scan MS spectra (350–1650 m/z) were acquired in Profile mode at a resolution of 60,000 at m/z 200, a maximum injection time of 20 ms and an AGC target value of $3 \times 10^6$ charges. Up to 10 most intense peptides per full scan were isolated using a 1.4 Th window and fragmented using higher energy collisional dissociation (normalized collision energy of 27). MS/MS spectra were acquired in centroid mode with a resolution of 30,000, a maximum injection time of 110 ms and an AGC target value of $1 \times 10^5$. Single charged ions, ions with a charge state above 5 and ions with unassigned charge states were not considered for fragmentation and dynamic exclusion was set to 20 s.

MS raw data processing was performed with MaxQuant (v 1.6.5.0) and its in-build label-free quantification algorithm MaxLFQ applying default parameters. Acquired spectra were searched against the human reference proteome (Taxonomy ID 9606) downloaded from UniProt (21-11-2018; 94731 sequences including isoforms) and a collection of common contaminants (244 entries) using the Andromeda search engine integrated in MaxQuant. FDR was set to 1% on protein, PSM and site decoy level. Statistical analysis was done with Perseus 2.0.7.0. Proteins were defined as interactors, if they passed a 5% FDR corrected one sided two-sample T-test with a minimal enrichment factor of two. The mass spectrometry proteomics data have been deposited to the ProteomeXchange Consortium (see Data Availability Statement).

## Phos-tagTM SDS-PAGE
Phos-tagTM acrylamide (Wako) gels were used as indicated by the supplier. Gels were prepared with 10% acrylamide, 50 μM phos-tagTM and 100 μM MnCl$_2$. Cells were lysed in SDS sample buffer supplemented with 10 μM MnCl$_2$.

## Proximity ligation assay (PLA)
HeLa cells were seeded onto black, clear flat bottom 96-well plates (2000 cells/well), treated with 10 ng/ml IFNγ for 24 h and infected with *Salmonella*. Cells were washed in PBS before fixation with 4% paraformaldehyde for 15 min at room temperature. Rabbit anti-IRGQ (HPA043254; Sigma) and mouse anti-HA (MMS-101P; Covance) were used with the respective Duolink in situ PLA probes (DUO92001, DUO92005; Sigma) and the PLA Duolink in situ detection reagent kit (DUO92008, Sigma) according to the manufacturer's instructions. After washes in PBS, cells were incubated with Hoechst 33342 and Alexa Fluor 647 Phalloidin for 1 h in the dark. Plates were imaged using a Yokogawa CQ1 microscope and quantification of the PLA signal was performed using the Yokogawa CQ1 software.

## Protein expression and purification
GST or His-tagged fusion proteins were expressed in E. coli strain BL21 (DE3). Bacteria were cultured in LB medium supplemented with 100 μg/mL ampicillin at 37 °C in a shaking incubator until OD600 ~0.5–0.6. Protein expression was induced by the addition of 0.5 mM IPTG and cells were incubated at 16 °C for 16 h. Bacteria were harvested by centrifugation ($3000 \times g$, 20 min) and lysed by sonication in GST lysis buffer (20 mM Tris HCl, pH 7.5, 10 mM EDTA, pH 8.0, 5 mM EGTA, 150 mM NaCl, 0.1% β-mercaptoethanol, 1 mM PMSF) or His lysis buffer (25 mM Tris HCl, pH 7.5, 200 mM NaCl, 0.1% β-mercaptoethanol, 1 mM PMSF, 1 mg/ml lysozyme). Lysates were cleared by centrifugation ($28000 \times g$), 0.05% of Triton X-100 was added and the lysates were incubated with glutathione Sepharose 4B beads (GE Life Sciences) or Ni-NTA agarose beads (Thermo Fisher) on a rotating platform at 4 °C

for 1 h. The beads were washed five times either in GST wash buffer (20 mM Tris HCl, pH 7.5, 10 mM EDTA, pH 8.0, 150 mM NaCl, 0.5% Triton X-100, 0.1% β-mercaptoethanol, 1 mM PMSF) or His wash buffer (25 mM Tris HCl, pH 7.5, 200 mM NaCl, 0.05% Triton X-100, 10 mM Imidazole). The immobilized proteins were reconstituted in GST storage buffer (20 mM Tris HCl, pH 7.5, 0.1% NaN3, 0.1% β-mercaptoethanol) or eluted with His elution buffer (25 mM Tris HCl, pH 7.5, 200 mM NaCl, 300 mM Imidazole) and dialyzed in (25 mM Tris HCl, pH 7.5, 200 mM NaCl) at 4 °C for 16 h. Recombinant GST-TBK1 was obtained from the MRC PPU DSTT in Dundee, UK (#DU12469)[18].

## FACS
T-REx mCherry-GFP GABARAPL2 WT, S10A and S10D mutant U2OS were cultivated in cell culture flasks until ~80% confluency and transferred to 6-well plates (500000 cells/well) for the experiments (3 replicates/condition for every cell line). Cells were treated overnight with doxycycline (final concentration: 1 μg/ml) to induce mCherry-GFP expression. Negative control was not induced. After induction, cells were washed once with PBS and treated as follows: untreated (stopped after 2 h), CCCP (40 μM, 2 h), EBSS (6 h) and EBSS + Bafilomycin (200 nM, 6 h). After treatments, cells were detached using trypsin/EDTA, centrifuged (5 min, $500 \times g$) and resuspended in 200 μl of FACS buffer (PBS + EDTA + FBS). FACS was performed directly at BD Symphony A5. Following singlet gating, cells were gated for high mCherry+ GFP+ cells using FlowJo software (version 10). The mean fluorescence intensity (MFI) for mCherry and GFP was calculated for each sample. Fold changes were calculated and normalized to the mean of each untreated cell line (WT, S10A, and S10D, respectively). Analysis was performed using GraphPad Prism software.

## Modeling and simulations of IRGQ-ATG8 complexes
Full-length sequences of IRGQ (UniProtKB: Q8WZA9), GABARAPL2 (UniProtKB: P60520), MLP3B (LC3B, UniProtKB: Q9GZQ8), and ATG7 (UniProtKB: O95352) were used to model the 3D structures of various complexes using AlphaFold2-multimer model (AF2) multimers: (1) IRGQ-GABARAPL2, (2) IRGQ-GABARAPL2-ATG7, and (3) IRGQ-GABARAPL2-LC3B-ATG7. We obtained 100 models for each complex using the default AFv2.0 parameters (database updated, 2025). To evaluate the stability of the IRGQ-GABARAPL2 complex, we performed all-atom molecular dynamics (MD) simulations using the PDB structure of the IRGQ$_{1-189}$-GABARAPL2 complex (PDB ID: 8Q6Q) as the initial model. Using the CHARMM-GUI server[36], we refined the native (WT) IRGQ-GABARAPL2 complex and also modeled the phosphorylated version (S10PO4). We ensured that the disulfide bridge between C152 and C158 was preserved in our models. Protein complexes were placed in an octahedral box and solvated with the TIP3P water model and physiological salt concentration (150 mM NaCl). All-atom MD simulations of the IRGQ-GABARAPL2 complexes were performed with GROMACS (v 2021.5)[37] using the CHARMM36m force field. Initially, the system was minimized using the steepest-descent algorithm until the maximum force reached 1000 kJ mol$^{-1}$ nm$^{-1}$. The equilibration phase was run in an NVT ensemble with the v-rescale thermostat at 310 K ($\tau_T = 1$ ps)[38]. Position restraints were applied to the backbone (400 kJ mol$^{-1}$ nm$^{-2}$) and sidechain (40 kJ mol$^{-1}$ nm$^{-2}$) atoms to equilibrate the water. During the production run, the pressure was maintained at 1 bar ($\tau_P = 5$ ps, compressibility = 4.5E-05 bar$^{-1}$) using an isotropic c-rescale barostat[39]. Three replicates of production runs were simulated in each condition for 1000 ns with a 2 fs timestep. Coarse-grained MD simulations for the IRGQ-GABARAPL2 complexes were performed using (WT) and a phosphomimetic variant (S10D). The Martini force field was used to map the atomistic structure of the top-ranked AF2 model with the martinize2.py script[40]. For both variants, we employed the Go-Martini 3.0 model. Secondary structure assignments were done using DSSP, followed by automatic identification of disulfide bonds. Backbone restraints were applied with a force

constant of 1000 kJ mol$^{-1}$ nm$^{-2}$, Go-like native contacts were modeled ($\varepsilon = 12$ kJ mol$^{-1}$, residue distance cutoff = 3Å), along with predefined intrinsically disordered regions (regions 1–7, 179–191, 331–430, and 617–623) explicitly treated to preserve their conformational flexibility. The CG models were placed in a hexagonal simulation box, solvated with coarse-grained water beads, with 0.15 M NaCl. MD simulations were performed using GROMACS (v 2021.5)[37]. Systems were first energy minimized for 3000 steps with the steepest-descent approach, followed by an equilibration in an NPT ensemble at 310 K. Temperature and pressure were maintained, respectively, with a v-rescale thermostat ($\tau_T = 1$ ps) and an isotropic c-rescale barostat ($\tau_P = 5$ ps, compressibility = 4.5E-05 bar$^{-1}$). During the production runs, we used the Parrinello-Rahman scheme ($\tau_P = 12$ ps)[41] to maintain system pressure. Simulations were performed with a 20 fs time step for 1000 ns, and 15 replicates were run for each system. Distances and contact maps for pairwise residue-residue interactions within each complex were computed using in-house scripts based on MDAnalysis v2.9.0[42]. Contact maps were computed by counting pairwise residue contacts between chain A and chain B according to $AB_{cnts} = [\sum_{i \in A} \sum_{j \in B} \sigma(|r_{ij}|)]$, where the sums extend over heavy atom positions of interacting residues (ij) and $\sigma(|r_{ij}|) = 1 - \left\lfloor 0.5 - 0.5(\tanh((|r_{ij}| - a)/b)) \right\rfloor$, a smooth sigmoidal counting function to limit interactions below the cut-off distance ($r_{ij} \leq a$), where the cutoff parameter, $a$ was set to 5 Å for atomistic simulations and 10 Å for CG simulations, while the smoothing parameter $b$ was set to 0.5 and 1.0, respectively. Binding free energies for the WT and S10PO4 variant complex from atomistic simulations were estimated using the MMPBSA approach as implemented in the gmx_MMPBSA program with default parameters[43]. The binding free energy was decomposed into gas-phase and solvation contributions, with the gas-phase term ($\Delta G_{gas}$) comprising van der Waals and electrostatic energies, and the solvation term ($\Delta G_{solv}$) including polar electrostatic contributions computed using the Poisson-Boltzmann model and nonpolar contributions estimated from solvent-accessible surface area. Per-residue free-energy decomposition was performed for both WT and S10PO4 systems to quantify residue-specific energetic contributions, and uncertainties were estimated from variability across independent replicas.

## Statistical analysis

All experiments have a minimum of three biological replicates. Data are presented as the mean with error bars indicating the s.d. (standard deviation). Statistical significance of differences between experimental groups was assessed with Student's $t$ test. Differences in means were considered significant if $p < 0.05$. Differences with $p < 0.05$ are annotated as *, $p < 0.01$ are annotated as ** and $p < 0.001$ are annotated as ***. All western blots shown are representative of biological replicates. Immunofluorescent images were analyzed with CellProfiler 4.2.8.

## Sequence alignments

Sequence alignments were performed using the Clustal algorithm[44] with Ensemble identifiers. This approach allowed for the rapid and accurate alignment of protein sequences from the Ensemble database, facilitating the identification of conserved motifs and regions of interest.

## Reporting summary

Further information on research design is available in the Nature Portfolio Reporting Summary linked to this article.

## Data availability

The data generated in this study are provided in the Supplementary Information/Source Data file. Source data are provided with this paper. Source data for Supplementary Figs. 1, 4, and 5 containing modeling and simulation data have been deposited in Zenodo repository (https://doi. org/10.5281/zenodo.19065624). Source Data for microscopy analysis, Fig. 5 and Supplementary Fig. 7 have also been deposited in Zenodo repository (https://doi.org/10.5281/zenodo.19008522). The mass spectrometry proteomic data have been deposited to the ProteomeXchange Consortium via the PRIDE partner repository with the following dataset identifiers: IRGQ-GABARAPL2 complex interactome data–PXD066671, GABARAPL2 interactome - PXD066665. Source data are provided with this paper.

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

## Acknowledgements

We thank Prof Kraiczy for the provided S2 Salmonella working space. We are also grateful to Henry Bailey for valuable discussions and providing critical data to our previous manuscript[16]. We acknowledge all members of the Quantitative Proteomics Unit at IBC2 (Goethe University, Frankfurt), in particular Thorsten Mosler, Florian Bonn and Georg Tascher, for support and expertise in proteomics methodology and data analysis, Martin Adrian-Allgood and Julia Pomirska for technical help and measurements, Kristina Wagner for preparing LC columns, and David Krause for help in (bio)informatics. We thank the FACS facility at the Helmholtz-Zentrum für Infektionsforschung GmbH, Braunschweig. We thank the Deutsche Forschungsgemeinschaft (German Research Foundation, DFG) for funding the liquid chromatography (LC)-MS system (easy nLC1200, Orbitrap Fusion LUMOS) used in this study (FuGG Project-ID: 403765277). The research was funded by Dr. Rolf M. Schweite Stiftung to I.D. (project 13/2017), grants from the Goethe University Frankfurt to L.H. (Nachwuchswissenschaftler grant 710000624), and GRADE A/B Focus to L.H. (PID003790), as well as the LOEWE Zentrum Frankfurt Cancer Institute Discovery & Development Grant to L.H. (21001366). S.A.P.-C., A.C., and R.M.B. thank the Center for Supercomputing, Goethe University Frankfurt (GUF), for computing time on the Goethe-HLR cluster. Additionally, this work was supported by the Clusterproject ENABLE funded by the Hessian Ministry for Science and the Arts to R.M.B, and L.H.; CRC project on selective autophagy, grant/award number: project-ID 259130777 to R.M.B, I.D. and L.H.; the Leistungszentrum Innovative Therapeutics (TheraNova) funded by the Fraunhofer Society and the Hessian Ministry of Science and Arts to I.D. and U.G.M. Work of the Herhaus lab at the Helmholtz-Zentrum für Infektionsforschung GmbH, Braunschweig is supported by the by the Microbial Stargazing program of the German Federal Ministry of Education and Research (BMFTR) under grant number 01KX2324. Responsibility for the content of this publication lies with the author.

## Author contributions

U.G.-M. designed/performed most of the experiments and analyzed the data. L.H. and I.D. conceived the study. S.A.P.-C. and A.C. performed molecular modeling and simulations with support and supervision from R.M.B. L.H., P.L, B.C., M.A., and A.V.A. performed some experiments and analyzed data. L.H. and I.D. managed the project and supervised experiments. U.G.-M. and L.H. wrote the manuscript with input from all authors.

## Funding

## Competing interests

The authors declare no competing interests.
