## [Transparent Peer Review file · Nature Communications]

TBK1 restricts IRGQ-mediated autophagy

Corresponding Author: Dr Lina Herhaus

Version 0:

Reviewer comments:

Reviewer #1

(Remarks to the Author)

The presented work follows on an earlier study of the Dikic/Herhaus labs on the identification of IRGQ as a novel autophagy cargo receptor. In that seminal work, IRGQ was identified as novel cargo receptor that recognizes misfolded MHC Class I and sends them for degradation through the autophagy pathway. In this work, the authors follow on that work by investigating the significance and regulation of the interaction between IRGQ and GABARAPL2, which occurs through a second LIR motif that provides an unusual specificity for GABARAPL2. Concomitant, in proteomics experiments GABARAPL2 appears as top hit for IRGQ, suggesting an important role for the interaction in the function or regulation of IRGQ's function. In this work, the authors present evidence for the GABARAPL2 residue S10 being a substrate for TBK1-phosphorylation. Sitting right at the interface with IRGQ, phosphorylation of this residue is suggested to regulate the interaction between IRGQ and GABARAPL2.

While this conceptual framework makes sense as the S10 sits indeed at the interface and introduction of a phospho-group may break the interaction interface, the presented results to support this model are not always equally convincing to this reviewer. Some experiments lack critical controls or the presented interpretation of the results could also be interpreted in a different way. Most importantly, the manuscript does not sufficiently address the fact that IRGQ contains two LIR motifs, which may explain several of the observed effects. For instance, somewhere in the manuscript it is suggested that GABARAPL2 binds to LC3B but without further biochemical data this conclusion is more likely to be explained by the fact that IRGQ has two LIR motifs and may thus bind both GABARAPL2 and LC3b at the same time – without there being a need for LC3B and GABARAPL2 to interact with each other directly. Below I will substantiate my comments in greater detail but much of the described effects or phenotypes may be attributed to IRGQ forming a cargo receptor with two LIR motifs and may therefore drive ATG8 lipidation, recruitment of the FIP200/ULK1 complex, etc. Secondly, if TBK1 would form the critical regulator of the IRGQ/GABARAPL2 interaction, then Figure 5 could be strengthened by testing TBK1 inhibitors effect on MHC I turnover. Third, some of the data (e.g. Fig 3D/E) appear difficult to reconcile with the stated conclusions, and may benefit from additional clarification. Fourth, a key question that remains unaddressed is why TBK1 would phosphorylate GABARAP S10 and when this regulation is important for IRGQ's function in the cell. For example, a lot of the experiments are done with the S10D mutant however this leads to a loss-of-interaction whereas the more relevant control (S10A) often shows no difference from the WT (e.g. Fig 5E) raising the question how important the proposed regulation truly is for IRGQ's role in the cell. So the S10D mutant clearly shows that the IRGQ/GABARAPL2 interaction is important, but lack of phenotypes for the S10A raise the question how important this phospho-regulation by TBK1 truly is.

Detailed comments:

- (1) Introduction: the authors write that the phagophore membrane arises from multiple sources within the cell. Perhaps a more accurate way of phrasing this is that the ATG9-vesicle forms the seed for the autophagosome and expands upon lipid influx from the ER into the growing phagophore through ATG2-proteins.
- (2) Introduction: the authors write that cargo selection is achieved through a LIR-motif which is presented by the cargo. While I may have misunderstood what was meant, it would be better to rephrase this more clearly such that the cargo receptors themselves contain a LIR-region required to interact with the autophagy machinery, as the cargo receptors typically recognize ubiquitin (in case of most soluble cargo receptors) or more specific substrate features such as non-conformational MHC class I molecules in case of IRGQ.
- (3) Results P5: it would be good to also add the AF scores as supplementary figure along with the color-coded predicted structure so that it is easier to read out where the flexible domains are located.
- (4) Results P6: the authors conclude based off Figure 1F that GABARAPL2 binds LC3B and that IRGQ regulates this interaction. Can the authors specify whether they think of this as a direct or indirect interaction? To me, the more likely

explanation here seems that IRGQ has the ability to bind both LC3B and GABARAPL2 given its two LIR-motifs, hence its upon its knockdown, LC3B no longer co-pellets with GABARAPL2-coated beads.

(5) Figure 2E: it would be good if the authors could also specify in the figure legend that the boxed cell is a non-transfected control cell for comparison to transfected cells (and not only in the main text).

(6) Figure 2: the authors conclude that the IRGQ/GABARAPL2 complex promotes ATG8 lipidation. However, an alternative interpretation of the presented results is that overexpressing IRGQ (just like any other cargo receptor) leads to ATG8 lipidation. So to rule out that the presented results are flawed by overexpressing IRGQ, the authors should perform additional control experiments where they overexpress LIR-deficient IRGQ and establish IRGQ/GABARAPL2 complex formation through artificial means (eg FRB/FKBP system). Additionally, single LIR mutants could provide further insights.

(7) The authors use the abbreviation LDS several times but do not specify its full name (LIR-docking site).

(8) Figure 3B: the authors use GST-TBK1 for their in vitro kinase assays. While the experiment seemed to have worked, tagging TBK1 at its N-terminus could affect substrate binding as the methionine 1 sits directly at the kinase domain and any N-terminal tag may thus interfere in unpredictable ways with substrate binding. Any revision experiments with recombinant TBK1 would therefore better be performed with untagged TBK1.

(9) Figure 3F: the authors conclude based on this figure that GABARAPL2-S10D loses interaction with IRGQ, ATG7, and ULK1. Based on this blot, only the interaction with IRGQ is lost whereas the interactions with ATG7 and ULK1 may vary slightly (hard to say without quantification) but are clearly not lost. This would also make sense as the mutated residue is likely on a surface area that is important for IRGQ binding but far away from the LDS.

(10) Results P8: the authors conclude based on Figure 2 and Suppl Figure S3 that phosphorylation of S10 affects the interaction of this protein to the autophagy initiation machinery. While it is unclear whether the authors consider IRGQ part of the autophagy initiation machinery, if not and they refer to the ULK1-complex and ATG7, then this conclusion does not stand with the presented data. Furthermore, given that IRGQ contains two LIR motifs, each IRGQ molecule can presumably bind one ATG8 molecule and one FIP200 molecule. This would also explain why the 10D mutant but not the 10A mutant causes loss of ULK1 and ATG13 binding due to the steric clash this substitution causes at the GABARAPL2/IRGQ interface.

(11) Figure S4C: this experiment would better be presented by using a more specific small molecule inhibitor for TBK1 (such as GSK8612) as MRT67307 cross-reacts with ULK1 and both kinases have been shown to have overlapping substrates. Also from the blot, there appears no difference between untreated versus MRT67307-treated cells in terms of p-TBK1 levels. The input is not shown so its hard to compare but based on the pull-down, there is an equal amount of p-TBK1 which would indicate that the drug has not worked, yet the authors see changes for the p-GABARAPL2 antibody.

(12) Results P8: This sentence references the wrong supplementary figure panel, this should be corrected into S4E: To more closely delineate the kinetics of this phosphorylation, we induced mitophagy and measured GABARAPL2 pS10 at various time points. GABARAPL2 pS10 peaks at 1h after treatment and coincides with maximal TBK1 activation (Fig. 4B, S4D).

(13) Results P8: same for figure S4D which should be S4E.

(14) While this reviewer would agree with the conclusion of Figure 4, that phosphorylation of residue S10 is likely to interfere with the interaction with IRGQ as this residue sits right at the interaction interface and any introduction of steric group or negative charge is likely to abrogate this interaction interface, the data supporting this conclusion are somewhat difficult to interpret (e.g. blot quality in Fig 4E) and in some cases appear not to fully align with the proposed model (e.g. Fig 4D). Figure 4D shows the largest reduction in IRGQ co-immunoprecipitation in the IFN γ treated cells but those cells show no activated p-TBK1 in the input. Instead, the two conditions with activated p-TBK1 detected in the input samples (both salmonella treated conditions) show only minor reduction in IRGQ pull-down by GABARAPL2.

(15) Figure 5B: the overall trends for the S10D appear the same as for the WT but the number of puncta per cell is just overall lower (even in the untreated condition). It would be helpful if the authors could clarify which control experiments were performed to exclude possible technical artifacts. The figure indicates that more than 200 cells were counted but from how many different transductions? The reason for this comment is because the authors wrote previously that TBK1 is not activated upon EBSS treatment, residue 10 is not phosphorylated under these conditions, hence one would not expect the S10D mutant to make a difference in this context. Yet, the authors see a difference and attribute importance to this observation.

Reviewer #2

(Remarks to the Author)

In this study, the authors discover a specific functional role for GABARAPL2 in the modulation of IRGQ turnover through its phosphorylation by the kinase TBK1. The authors also discover the phospho site and show that phosphorylation of GABARAPL2 at S10 by TBK1 releases it from IRGQ and prevents the formation of autophagy initiation complexes. This was shown to have an effect on surface HLA turnover activity by IRGQ. Overall, this is a very interesting study. There are six ATG8 family members and while broad roles have been assigned to LC3 and GABARAP subfamily members, specific roles for ATG8 proteins are lacking, making the discovery here regarding GABARAPL2 one of the few reports to do so. In addition, TBK1 is largely a pro-selective autophagy kinase, but the authors show a negative regulation role for TBK1 in selective autophagy that is quite interesting. Given that TBK1 is strongly activated during selective autophagy of mitochondria and Salmonella, it would be beneficial for the authors to explore whether TBK1 activation to turn off IRGQ turnover is part of its role in enabling mitophagy or xenophagy, potentially by redistributing autophagy initiation complexes away from IRGQ and

towards mitochondria or bacteria. Nevertheless, the study is of broad significance and interest, and the data are clear and largely convincing. Some specific suggestions are included below.

Major:

1. Figures 1 and 2: Is the Venus (BiCap) system binding between IRGQ and GABARAPL2 dependent on the LIR1 motif within IRGQ? This is a helpful control to clarify whether the Venus system binding experiments are largely driven between LIR mediated interactions or the Venus coming together. In addition, is the delivery to lysosomes of the Venus complex dependent on LIR1?
2. Figure 2D: The authors conclude that increased LC3B lipidation and increased LC3B foci correlates with increased autophagy. However, a block or decrease in autophagy flux can produce similar results. can the authors assess autophagy flux, perhaps through analysing p62 turnover, to assess whether autophagy flux is affected in cells with or without Venus system overexpression?.
3. Figure 4C and S4G: Can the authors provide example images of the PLA experiments?
4. After CCCP treatment, GABARAPL2 flux was considerably lower in WT and S10D cells compared to untreated conditions. Importantly, there was no difference between the two cell lines (Fig. S5B), confirming that TBK1 restricts the IRGQ-GABARAPL2 autophagy axis.
Based on the above, what effect does the S10A mutant have on GABARAPL2 flux?
5. In all experiments with CCCP treatment, are the cell lines also expressing Parkin? How was Parkin expression introduced? This seems to be missing from the methods.
6. Figure 5B: Can the authors assess if other autophagy substrates/pathways are affected by S10D? For example, turnover of an ERphagy receptor, or p62 (with EBSS treatment), or alternatively if PINK1 Parkin mitophagy or Salmonella clearance is affected by the S10D mutation? This will help to clarify what effect S10D (TBK1 phosphorylation of GABARAPL2) might have on the activity of parallel selective autophagy pathways once IRGQ turnover is shut down.

Minor

1. "TBK1 phosphorylates members of the LC3 family on several residues"
Given that the authors go on to refer to GABARAPL2, it would be more accurate to state ATG8 family, LC3 is a subfamily of the ATG8s.

Version 1:

Reviewer comments:

Reviewer #1

(Remarks to the Author)

The comments and concerns have been adequately addressed. No further comments.

Reviewer #2

(Remarks to the Author)

The authors have satisfactorily addressed the comments. Congratulations on an interesting discovery.

Point-by-point letter NCOMMS-25-60288: TBK1 restricts IRGQ-mediated autophagy

Reviewer #1 (Remarks to the Author):

The presented work follows on an earlier study of the Dikic/Herhaus labs on the identification of IRGQ as a novel autophagy cargo receptor. In that seminal work, IRGQ was identified as novel cargo receptor that recognizes misfolded MHC Class I and sends them for degradation through the autophagy pathway. In this work, the authors follow on that work by investigating the significance and regulation of the interaction between IRGQ and GABARAPL2, which occurs through a second LIR motif that provides an unusual specificity for GABARAPL2. Concomitant, in proteomics experiments GABARAPL2 appears as top hit for IRGQ, suggesting an important role for the interaction in the function or regulation of IRGQ's function. In this work, the authors present evidence for the GABARAPL2 residue S10 being a substrate for TBK1-phosphorylation. Sitting right at the interface with IRGQ, phosphorylation of this residue is suggested to regulate the interaction between IRGQ and GABARAPL2.

While this conceptual framework makes sense as the S10 sits indeed at the interface and introduction of a phospho-group may break the interaction interface, the presented results to support this model are not always equally convincing to this reviewer. Some experiments lack critical controls or the presented interpretation of the results could also be interpreted in a different way. Most importantly, the manuscript does not sufficiently address the fact that IRGQ contains two LIR motifs, which may explain several of the observed effects. For instance, somewhere in the manuscript it is suggested that GABARAPL2 binds to LC3B but without further biochemical data this conclusion is more likely to be explained by the fact that IRGQ has two LIR motifs and may thus bind both GABARAPL2 and LC3b at the same time – without there being a need for LC3B and GABARAPL2 to interact with each other directly. Below I will substantiate my comments in greater detail but much of the described effects or phenotypes may be attributed to IRGQ forming a cargo receptor with two LIR motifs and may therefore drive ATG8 lipidation, recruitment of the FIP200/ULK1 complex, etc. Secondly, if TBK1 would form the critical regulator of the IRGQ/GABARAPL2 interaction, then Figure 5 could be strengthened by testing TBK1 inhibitors effect on MHC I turnover. Third, some of the data (e.g. Fig 3D/E) appear difficult to reconcile with the stated conclusions, and may benefit from additional clarification. Fourth, a key question that remains unaddressed is why TBK1 would phosphorylate GABARAP S10 and when this regulation is important for IRGQ's function in the cell. For example, a lot of the experiments are done with the S10D mutant however this leads to a loss-of-interaction whereas the more relevant control (S10A) often shows no difference from the WT (e.g. Fig 5E) raising the question how important the proposed regulation truly is for IRGQ's role in the cell. So the S10D mutant clearly shows that the IRGQ/GABARAPL2 interaction is important, but lack of phenotypes for the S10A raise the question how important this phospho-regulation by TBK1 truly is.

We would like to sincerely thank the reviewer for their very positive assessment of our manuscript and for the thoughtful, constructive suggestions that have helped us significantly improve the clarity and strength of our study. We are grateful that the

reviewer appreciates both the conceptual framework and the broader significance of our work, following up on the initial discovery of IRGQ as a cargo receptor. In response to the reviewer's valuable comments, we have now incorporated several key additions and clarifications:

1. TBK1 inhibition and MHC-I turnover: As suggested, we performed additional experiments to directly test whether TBK1 activity influences IRGQ function. We now include a new figure (5H) assessing the effect of TBK1 inhibitors on MHC-I turnover. These data further support the regulatory role of TBK1-mediated GABARAPL2 phosphorylation in IRGQ-dependent cargo degradation.

2. Molecular modelling of phosphorylated GABARAPL2 complexes: To provide a mechanistic and structural basis of our findings, we performed molecular modeling and molecular dynamics simulations of phosphorylated GABARAPL2 in complex with components of the autophagy machinery. This analysis further illustrates how modification of S10 may selectively impair IRGQ binding while sparing LDS-mediated interactions with other ATG8 interactors (new Figures S4A-H).

We have updated the manuscript text accordingly: To provide additional support for how TBK1-dependent phosphorylation modulates this selective interaction, we performed molecular modelling and molecular dynamics simulations of phosphorylated GABARAPL2 in the context of the IRGQ-ATG8 interface and additional ATG8-dependent assemblies. These models position Ser10 within the IRGQ-binding surface

of GABARAPL2, such that introduction of a phosphate group (or the S10D phosphomimetic) perturbs the local electrostatic and steric environment at the IRGQ contact site, thereby weakening IRGQ engagement. Importantly, Ser10 is spatially separated from the canonical LDS, and our modelling is consistent with the experimental observation that LDS-mediated interactions with other ATG8 interactors are largely preserved. Together, these structural analyses (Fig. S4A-H) strengthen the rationale for a mechanism in which TBK1 phosphorylation selectively destabilizes the IRGQ–GABARAPL2 complex while sparing broader ATG8 interactomes.

Figure R2: X-ray model (8Q6Q) of the N-terminal domain (pink) of IRGQ containing LIR1 (186–189, cyan) in complex with GABARAPL2 (grey). Helix 2 of GBRL2 (purple) makes contacts within a pocket formed by IRGQ switch I (partially disordered), switch II, and the linker spanning LIR1 site and the N-terminal G-domain (right, zoom-up). Key ionic interactions across this interface (> 10 residue pairwise contacts) between IRGQ₁₋₁₈₆ (yellow) and GBRL2 (purple) contribute to the complex stability.

Figure R3: Phosphorylation of GABARAPL2 destabilizes the IRGQ–GBRL2 switch I and II interface. (A) Time series of distances between interacting residue pairs estimated from atomistic MD simulations (3 replicates x 1000 ns) of WT complex (left), and the phosphorylated complex (middle, S10–PO4). Comparison of density distributions of distances (WT vs. S10–PO4) across select interface residue-pairs (top to bottom) for E74–S10(OG), W71(NE1)–S10(OG), F57(CZ)–K20(NZ), and

L35(CD1)–A23(CB). (right) Selected snapshots (indicated times) corresponding of the complex simulations showing the structural rearrangements at the interface for S10–PO4 indicating destabilization. Distances are shorter in WT complex in comparison to S10–PO4. (B) MMPBSA energy computation and its decomposition into various terms shows relative contributions to the total binding energy of the IRGQ–GBRL2 complex (WT, blue; S10PO4, green). S10–PO4 complex displays altered balance between electrostatic and solvation components, consistent with the structural reorganization of the interface induced by excess negative charge. (C) Residue-wise energy decomposition shows destabilizing effects for E74 (IRGQ) and S10 (GBRL2) only in the S10–PO4 variant, due to electrostatic repulsion across the interface.

Figure R4: Representative snapshots from coarse-grained Go-Martini 3.0 simulations of IRGQ–GABARAPL2 complex (WT vs. S10D). (A) Representative snapshot from WT complex simulations (15 replicates, 1000 ns each) showing preservation of the native interface. The LIR1 region remains

stably engaged with GABARAPL2, and key interfacial residues (including K24, K20, A19, S10, and W71) maintain persistent contacts (right, zoom-up). **(B)** By contrast, the S10D complex exhibits a disrupted native interface. The presumed interface residues mediate interactions with alternative sites on GABARAPL2, causing instability. **(C)** Distributions of # of interface contacts in WT (blue) exhibits consistently higher interactions in comparison to S10D (green) **(D)**. Heatmap showing the difference in pairwise residue–residue contact probability between WT and S10D ($\Delta\text{cnts.} = C_{ij}^{\text{S10D}} - C_{ij}^{\text{WT}}$). Most interface residue pairs strongly interact in the WT complex in comparison to S10D, consistent with altered electrostatic effects even in CGMD runs.

3. Clarification of IRGQ’s dual LIR motifs and their functional implications: Following the reviewer’s important point, we adjusted the text to more explicitly describe the role of IRGQ’s two LIR motifs and how they explain several observed interactions, including simultaneous engagement of GABARAPL2 and LC3B. We now clarify how this dual-LIR architecture shapes IRGQ’s activity as a cargo receptor and how it integrates into our proposed regulatory model.

4. Clarification of data interpretation (e.g., Fig. 3D/E): We carefully revised the text to ensure that the conclusions drawn from these experiments more precisely reflect the data and can not be misunderstood again.

5. Relevance of S10 phosphorylation and the role of S10A: We thank the reviewer for raising this important point. We now specifically clarify that TBK1-mediated phosphorylation of GABARAPL2 at S10 functions as a negative regulatory termination signal. Specifically, our model proposes that phosphorylation at S10 terminates IRGQ–GABARAPL2 interactions and prevents further trafficking of GABARAPL2 to lysosomes. In this framework, phosphorylation is transient and conditional, occurring only once the relevant step of the pathway has been completed. This mechanistic interpretation explains why the S10A mutant closely phenocopies WT in many assays. Under basal conditions, GABARAPL2 is predominantly unphosphorylated, and therefore WT and S10A behave similarly. In contrast, the S10D mutant mimics constitutive phosphorylation, enforcing a permanent “off” state that blocks IRGQ binding and downstream function, thereby revealing the importance of regulated disengagement rather than continuous interaction. Importantly, we have now added new data directly comparing WT, S10A, and S10D mutants, which strengthens this conclusion (new Figure S7A). These experiments demonstrate that S10D specifically disrupts IRGQ binding and lysosomal targeting, while WT and S10A support normal progression through the pathway, consistent with phosphorylation acting as a termination mechanism. This figure has been added to the manuscript as Figure S7A and replaces old Figure S5B, which was not normalised and missing the controls, as pointed out by both reviewers. Together, these findings support a model in which TBK1-mediated phosphorylation of GABARAPL2 is critical for timely disengagement and pathway resolution, rather than for basal IRGQ function, resolving the apparent lack of phenotype observed with the S10A mutant. We expanded the discussion to address why physiological phosphorylation of GABARAPL2-S10 may be context-dependent and why the S10A mutant may not exhibit phenotypes under the conditions examined.

Figure R5: GABARAPL2-to-lysosome flux under stress is lost for the S10D phosphorylation mutant. Quantification of GABARAPL2-to-lysosome flux in cells expressing GABARAPL2 WT, S10A, or S10D. Each point represents the mean mCherry/GFP-mCherry puncta ratio per well and per biological replicate (four independent experiments, two wells per condition). Data were first normalized to the mean untreated (UT) signal averaged across WT, S10A, and S10D within each replicate, and then rescaled so that UT = 1 for all genotypes in the final plot. Conditions correspond to nutrient starvation (EBSS; 1–4 h), mitochondrial depolarization (40 μ M CCCP; 1–4 h), or lysosomal inhibition (200 nM BAF for 4 h; \pm EBSS or CCCP). One-sample Wilcoxon signed-rank with Holm p-value correction. **** $p < 0.0001$; *** $p < 0.001$; ** $p < 0.01$; * $p < 0.05$; n.s., not significant.

We are grateful for the reviewer’s insightful remarks, which have greatly strengthened the mechanistic clarity and overall impact of our manuscript. We hope that the revisions fully address the concerns raised and further highlight the importance of the IRGQ–GABARAPL2 axis in autophagy regulation.

Detailed comments:

(1) Introduction: the authors write that the phagophore membrane arises from multiple sources within the cell. Perhaps a more accurate way of phrasing this is that the ATG9-vesicle forms the seed for the autophagosome and expands upon lipid influx from the ER into the growing phagophore through ATG2-proteins.

We thank the reviewer for pointing this out and have expanded the text as follows: The phagophore membrane arises from ATG9-positive vesicles, which serve as the initial membrane seed. ATG2-mediated lipid transport from the endoplasmic reticulum (ER) is required for expansion and subsequent phagophore formation and is governed by a multiprotein complex composed of ULK1/2, ATG13, FIP200, and ATG101

(2) Introduction: the authors write that cargo selection is achieved through a LIR-motif which is presented by the cargo. While I may have misunderstood what was meant, it would be better to rephrase this more clearly such that the cargo receptors themselves contain a LIR-region required to interact with the autophagy machinery, as the cargo

receptors typically recognize ubiquitin (in case of most soluble cargo receptors) or more specific substrate features such as non-conformational MHC class I molecules in case of IRGQ.

We thank the reviewer for pointing this out and have expanded the text as follows: Cargo selection in these pathways is often achieved via the binding of an autophagy receptor that harbors a consensus LC3 interacting region (LIR), a stretch of 4 amino acids to facilitate its direct binding to the LDS (LIR-docking site) of LC3 orthologs.

(3) Results P5: it would be good to also add the AF scores as supplementary figure along with the color-coded predicted structure so that it is easier to read out where the flexible domains are located.

We thank the reviewer for this helpful suggestion. We have now included the AlphaFold confidence metrics as a new Supplementary Figure S1C, showing the color-coded predicted structure (pLDDT) alongside the corresponding predicted aligned error (PAE)/AF confidence scores. This addition makes it straightforward to identify flexible/low-confidence regions and interpret domain organization in the model.

Figure R6: Top-ranked AlphaFold2 model for the IRGQ-GABARAPL2 complex. Confidence on local predicted structure (pLDDT) is averaged over the top 100 models and mapped onto the structure (rainbow). GABARAPL2 is represented as a gray surface. Arrows indicate three key regions of IRGQ: LIR1 (cyan), LIR2 (pink), and switch II (green). Bottom panels show zoom-up of the LIR-LDS (left) and switch I/II protein-protein interfaces (right, $\langle pLDDT \rangle > 80$).

(4) Results P6: the authors conclude based off Figure 1F that GABARAPL2 binds LC3B and that IRGQ regulates this interaction. Can the authors specify whether they think of this as a direct or indirect interaction? To me, the more likely explanation here seems that IRGQ has the ability to bind both LC3B and GABARAPL2 given its two LIR-motifs, hence its upon its knockdown, LC3B no longer co-pellets with GABARAPL2-coated beads.

Yes, this is correct. We apologize for the confusion and rephrased the text as follows: Due to both LIR motifs, IRGQ affects the co-precipitation of GABARAPL2, LC3B, and the autophagy initiation machinery.

(5) Figure 2E: it would be good if the authors could also specify in the figure legend that the boxed cell is a non-transfected control cell for comparison to transfected cells (and not only in the main text).

The following sentence has been added to the figure legend: The boxed cells are non-transfected control cells for comparison to the transfected cells.

(6) Figure 2: the authors conclude that the IRGQ/GABARAPL2 complex promotes ATG8 lipidation. However, an alternative interpretation of the presented results is that overexpressing IRGQ (just like any other cargo receptor) leads to ATG8 lipidation. So to rule out that the presented results are flawed by overexpressing IRGQ, the authors should perform additional control experiments where they overexpress LIR-deficient IRGQ and establish IRGQ/GABARAPL2 complex formation through artificial means (eg FRB/FKBP system). Additionally, single LIR mutants could provide further insights.

We thank the reviewer for pointing out this concern that increased ATG8 lipidation could result from IRGQ overexpression per se. When this experiment was performed, we had already included additional IRGQ mutants targeting the N-terminal region that mediates interaction with GABARAPL2. Importantly, these mutants fail to promote GABARAPL2 lipidation despite comparable expression levels, demonstrating that lipidation depends on the integrity of the IRGQ–GABARAPL2 interaction rather than on IRGQ overexpression alone. We have updated the manuscript accordingly.

Figure R7: SDS-PAGE and Western blot of endogenously tagged HA-GABARAPL2 HeLa cell lysates with transfected Flag-IRGQ WT or mutants, treated with EBSS, 200 nM Bafilomycin A1 (3 h) or treated for CCCP 40 μ M (3h).

(7) The authors use the abbreviation LDS several times but do not specify its full name (LIR-docking site).

We thank the reviewer for pointing this out and have now included this explanation in the introduction.

(8) Figure 3B: the authors use GST-TBK1 for their in vitro kinase assays. While the experiment seemed to have worked, tagging TBK1 at its N-terminus could affect substrate binding as the methionine 1 sits directly at the kinase domain and any N-terminal tag may thus interfere in unpredictable ways with substrate binding. Any revision experiments with recombinant TBK1 would therefore better be performed with untagged TBK1.

We agree with the reviewer that it would be preferable to work with an untagged version of TBK1; however, in this case, the GST-tag did not impede TBK1's ability to phosphorylate GABARAPL2. Similarly, other studies have used the GST-tagged version of TBK1 for experiments (cf Figure 4D: <https://pmc.ncbi.nlm.nih.gov/articles/PMC2682862/>)

(9) Figure 3F: the authors conclude based on this figure that GABARAPL2-S10D loses interaction with IRGQ, ATG7, and ULK1. Based on this blot, only the interaction with IRGQ is lost whereas the interactions with ATG7 and ULK1 may vary slightly (hard to say without quantification) but are clearly not lost. This would also make sense as the mutated residue is likely on a surface area that is important for IRGQ binding but far away from the LDS.

We apologize for the confusion, as we agree with the reviewer that the GABARAPL2-S10D mutant does not fully lose interaction with ATG7 or ULK1. We conclude that the S10D mutation most prominently disrupts the interaction with IRGQ, while interactions with ATG7 and ULK1 are modestly affected. This is consistent with the structural prediction that Ser10 lies within a surface region important for IRGQ binding but is spatially separated from the LDS, which mediates ATG7 and ULK1 engagement. We have revised the text accordingly: HA-GABARAPL2 WT interacts with IRGQ, ATG7, and ULK1 under both basal and starvation conditions, whereas the HA-GABARAPL2 S10D mutant shows a pronounced reduction in IRGQ binding, with modest effects on ATG7 and ULK1 interactions (Fig. 3F). This selective loss of IRGQ engagement is consistent with our interactome analysis (Fig. 3E) and with the reduced association of these factors observed upon IRGQ knockdown (Fig. 1G). Autophagy-related proteins such as ULK1 typically bind ATG8 family members through LIR motifs that interact with the LDS. Ser10 in GABARAPL2 is located at a considerable distance from the LDS (Fig. S3H) and its phosphorylation selectively impairs IRGQ binding. Because IRGQ likely stabilizes or promotes the assembly of complexes with ATG7 and ULK1, the weakened IRGQ–GABARAPL2 interaction in the S10D mutant may indirectly account for the subtle changes observed with these autophagy-initiation factors. This supports a model in which IRGQ functions as an interaction hub coordinating GABARAPL2 with components of the autophagy-initiation machinery.

(10) Results P8: the authors conclude based on Figure 2 and Suppl Figure S3 that phosphorylation of S10 affects the interaction of this protein to the autophagy initiation machinery. While it is unclear whether the authors consider IRGQ part of the autophagy initiation machinery, if not and they refer to the ULK1-complex and ATG7, then this conclusion does not stand with the presented data. Furthermore, given that

IRGQ contains two LIR motifs, each IRGQ molecule can presumably bind one ATG8 molecule and one FIP200 molecule. This would also explain why the 10D mutant but not the 10A mutant causes loss of ULK1 and ATG13 binding due to the steric clash this substitution causes at the GABARAPL2/IRGQ interface.

We thank the reviewer for this careful critique and agree that our wording was too broad. In the revised manuscript, we have clarified that phosphorylation of GABARAPL2 at S10 primarily impairs binding to IRGQ, and that any effects on recruitment of ULK1-complex components or ATG7 are indirect rather than reflecting a direct loss of LDS-mediated interactions. First, we do not consider IRGQ itself to be part of the canonical autophagy initiation machinery (ULK1/FIP200/ATG13/ATG101). Rather, our data support a model in which IRGQ functions as an adaptor/cargo receptor that couples ATG8s to upstream factors. Consistent with the reviewer's point, the blots do not support a complete loss of ULK1-complex/ATG7 binding to GABARAPL2-S10D; instead, we observe the most pronounced loss for IRGQ, while ULK1/ATG7 show at modest changes. We have revised the text accordingly to avoid overinterpretation. Second, we explicitly address the dual-LIR architecture of IRGQ. Our AlphaFold-based modelling (new Figures S5A, B) supports that a single IRGQ molecule can engage two ATG8 family members via distinct regions: the N-terminal site preferentially binds GABARAPL2, whereas the C-terminal site is compatible with LC3B binding. In this framework, IRGQ can act as a scaffold that brings GABARAPL2 and LC3B into proximity, thereby facilitating co-recruitment of downstream factors that interact with ATG8s (including ATG7 and ULK1-complex components) without requiring a direct IRGQ–ATG7 interaction. We therefore interpret the reduced ULK1 association seen with S10D as a secondary consequence of disrupting the IRGQ–GABARAPL2 interface (and thus destabilizing the larger assembly), rather than as evidence that S10 phosphorylation directly regulates LDS-dependent binding. Accordingly, we have updated the Results and Discussion to state that S10 phosphorylation selectively disrupts the IRGQ–GABARAPL2 interface, and that IRGQ likely promotes proximity/assembly of GABARAPL2, LC3B, and their respective interactors, explaining the observed effects on the initiation machinery readouts.

Figure R8: AF2 multimers show differential IRGQ-hATG8 binding modes. (A) Predicted IRGQ-GBRL2-ATG7 complexes display ATG7 C-terminal interacting with GBRL2 in > 70% of the models (top left), while < 20% of the models represent IRGQ and ATG7 interacting with GBRL2 using different interfaces (top right). Bottom panels display the pairwise interfaces (interfacing regions are coloured in

yellow and violet for the first and second protein). (B) Predicted IRGQ-GBRL2-LC3B-ATG7 complexes display > 70% of models in which ATG7 interacts with GBRL2 and IRGQ interacts with LC3B (top left), while < 20% of the models present exchanged interactions, with ATG7 interacting with LC3B and IRGQ interacting with GBRL2 (top right). Bottom panels display the pairwise interfaces (interfacing regions are coloured in yellow and violet for the first and second protein).

(11) Figure S4C: this experiment would better be presented by using a more specific small molecule inhibitor for TBK1 (such as GSK8612) as MRT67307 cross-reacts with ULK1 and both kinases have been shown to have overlapping substrates. Also from the blot, there appears no difference between untreated versus MRT67307-treated cells in terms of p-TBK1 levels. The input is not shown so its hard to compare but based on the pull-down, there is an equal amount of p-TBK1 which would indicate that the drug has not worked, yet the authors see changes for the p-GABARAPL2 antibody.

We agree that, in principle, using the most selective TBK1 inhibitor available is desirable. However, the MRT67307 experiment was performed before the publication and broad availability of the highly selective TBK1 probe GSK8612 (reported in 2019). <https://pubmed.ncbi.nlm.nih.gov/31097999/>

To address the concern about MRT67307's known off-target activity and the possibility of overlapping substrate specificities, we obtained an additional, chemically distinct TBK1 inhibitor: BX795, which likewise suppresses the phosphorylation signal detected by the p-GABARAPL2 antibody (added to the revised manuscript as Figure S6C). <https://pmc.ncbi.nlm.nih.gov/articles/PMC2682862/>

Figure R9: SDS-PAGE and Western blot of HA-IPs from endogenously tagged HA-GABARAPL2 WT HeLa cells after treatment with TBK1 inhibitors (MRT67307 and BX795) or siRNA knock-down, plus CCCP treatment (1h).

Regarding the observation that p-TBK1 (Ser172) appears similar in untreated versus MRT67307-treated samples: Ser172 phosphorylation is not necessarily a reliable pharmacodynamic marker for TBK1 catalytic inhibition in cells. Indeed, it has been shown that while BX795 blocks TBK1/IKK ϵ downstream signaling (e.g., substrate phosphorylation), it does not inhibit—and can even enhance—Ser172 phosphorylation of endogenous TBK1/IKK ϵ , consistent with Ser172 phosphorylation being maintained via upstream kinase activity/feedback rather than strictly reflecting ongoing substrate phosphorylation by TBK1. <https://pmc.ncbi.nlm.nih.gov/articles/PMC2682862/> and <https://www.biorxiv.org/content/10.1101/2022.10.11.511671v1.full.pdf>

Therefore, the unchanged p-TBK1 (Ser172) band does not indicate inhibitor failure; instead, we interpret inhibitor efficacy based on suppression of phosphorylation of

bona fide downstream targets (i.e., GABARAPL2), which is the relevant readout in this experiment.

(12) Results P8: This sentence references the wrong supplementary figure panel, this should be corrected into S4E: To more closely delineate the kinetics of this phosphorylation, we induced mitophagy and measured GABARAPL2 pS10 at various time points. GABARAPL2 pS10 peaks at 1h after treatment and coincides with maximal TBK1 activation (Fig. 4B, S4D).

We thank the reviewer for pointing this out and have corrected the text.

(13) Results P8: same for figure S4D which should be S4E.

We thank the reviewer for pointing this out and have corrected the text.

(14) While this reviewer would agree with the conclusion of Figure 4, that phosphorylation of residue S10 is likely to interfere with the interaction with IRGQ as this residue sits right at the interaction interface and any introduction of steric group or negative charge is likely to abrogate this interaction interface, the data supporting this conclusion are somewhat difficult to interpret (e.g. blot quality in Fig 4E) and in some cases appear not to fully align with the proposed model (e.g. Fig 4D). Figure 4D shows the largest reduction in IRGQ co-immunoprecipitation in the IFN γ treated cells but those cells show no activated p-TBK1 in the input. Instead, the two conditions with activated p-TBK1 detected in the input samples (both salmonella treated conditions) show only minor reduction in IRGQ pull-down by GABARAPL2.

We agree that, structurally, S10 sits at the IRGQ–GABARAPL2 interface and that introducing a bulky phosphate/negative charge is expected to weaken binding. The intent of Figures 3 and 4 is to provide biochemical support for this model. We now explicitly point the reader to the orthogonal evidence throughout the manuscript that S10 phosphorylation (or phosphomimic) is sufficient to abrogate IRGQ binding (Fig. 3D–F, Fig. S3E–F, Fig. 4C, Fig. S6J, K), which is fully consistent with the proposed interface model.

In Figure 4D the key issue is kinetic and mechanistic uncoupling between (a) what is detectable as bulk TBK1 Ser172 phosphorylation at the specific harvest time and (b) the cumulative functional consequence of TBK1 activity on downstream substrates over a long stimulation window. TBK1 activation-loop phosphorylation (Ser172) is often transient and tightly feedback-controlled, such that it can peak at earlier time points and subsequently return toward baseline, even though downstream substrate phosphorylation and phenotypic effects have already occurred and persist. This is well documented for TBK1/IKK ϵ signaling, where activation is regulated by upstream inputs and feedback loops that limit hyperactivation. <https://pmc.ncbi.nlm.nih.gov/articles/PMC2682862/>

Moreover, pSer172 is not a straightforward “activity meter” for downstream phosphorylation in cell lysates: it can be maintained by upstream kinase activity/feedback and does not necessarily scale with net substrate phosphorylation measured later. <https://pmc.ncbi.nlm.nih.gov/articles/PMC2682862/>

Consistent with this principle, interferon-driven TBK1 pathway activation can be observed as an early time-dependent event (minutes-to-hours) rather than a 24 h steady-state readout, whereas the downstream transcriptional and post-translational consequences accumulate across longer exposures.

<https://www.frontiersin.org/journals/pharmacology/articles/10.3389/fphar.2022.987979/full>

In our experiment, IFN γ treatment was performed for 24 h, i.e., a time frame designed to robustly establish the IFN-stimulated state and allow the downstream consequences to manifest. TBK1/IKK ϵ have been implicated in IFN γ responses via IRF circuitry (including IRF7-dependent programs), supporting the concept that TBK1 engagement can occur during IFN γ conditioning even if pSer172 is not maximal at the terminal 24 h harvest. <https://pmc.ncbi.nlm.nih.gov/articles/PMC3295005/>

Hence, in Figure 4D, IRGQ dissociation reflects the accumulated displacement driven by S10 phosphorylation of GABARAPL2 over the IFN γ conditioning period, while the p-TBK1 (Ser172) input is only a snapshot at the endpoint and can be low/undetectable at 24 h despite earlier pathway engagement.

The Salmonella conditions were harvested to capture an infection-associated signaling state in which TBK1 activation-loop phosphorylation can be readily detected at that snapshot. However, at that time point, only a fraction of cellular GABARAPL2 is expected to have undergone S10 phosphorylation (and/or only a subset of complexes is remodeled), yielding a correspondingly more modest net change in co-IP. This interpretation is reinforced by the broader dataset in the manuscript showing that direct S10 phosphorylation (pS10) or phosphomimic (S10D) is sufficient to abolish IRGQ binding (Fig. 3D–F, Fig. S3E–F, Fig. 4C, Fig. S6J, K), supporting the mechanistic model independent of endpoint p-TBK1 intensity.

Overall, we believe the data support the conclusion that S10 phosphorylation of GABARAPL2 disrupts IRGQ binding, and that the apparent endpoint p-TBK1 discrepancy in Fig. 4D is explained by the known dynamics and interpretational limitations of pSer172 as a bulk readout at a late (24 h) time point.

(15) Figure 5B: the overall trends for the S10D appear the same as for the WT, but the number of puncta per cell is just overall lower (even in the untreated condition). It would be helpful if the authors could clarify which control experiments were performed to exclude possible technical artifacts. The figure indicates that more than 200 cells were counted but from how many different transductions? The reason for this comment is because the authors wrote previously that TBK1 is not activated upon EBSS treatment, residue 10 is not phosphorylated under these conditions, hence one would not expect the S10D mutant to make a difference in this context. Yet, the authors see a difference and attribute importance to this observation.

We thank the reviewer for this careful assessment of Fig. 5B. We agree that TBK1 is not activated upon EBSS treatment and that Ser10 is not phosphorylated under these conditions. The quantification shown in Fig. 5B was derived from three independent biological replicates. For each replicate, more than 200 cells in total were analyzed across two technical wells per condition. The reduced number of puncta observed in GABARAPL2-S10D-expressing cells, including under untreated conditions, likely reflects the constitutive inability of the phosphomimetic mutant to bind IRGQ. As a consequence, IRGQ-dependent stabilization of ATG8-positive structures is impaired, resulting in an overall reduced capacity for LC3B flux, independent of acute autophagy induction. Importantly, the relative response to starvation remains comparable between WT and S10D, consistent with the absence of TBK1 activation under EBSS conditions. To address the reviewer's concern, we repeated the experiment and performed a more comprehensive analysis of LC3-to-lysosome flux under multiple stress conditions. We now include quantification of LC3-to-lysosome flux in cells expressing GABARAPL2 WT, S10A, or S10D. Each data point represents the normalized mean ratio per well and per biological replicate (three independent experiments, two wells per condition), normalized to the untreated WT mean within each repeat. Conditions included starvation (EBSS, 1–4 h), mitochondrial depolarization (40 μ M CCCP, 1–4 h), and lysosomal inhibition (200 nM BAF for 4 h, \pm EBSS or CCCP). These additional data demonstrate that LC3-to-lysosome flux under stress conditions is not significantly altered by changes in GABARAPL2-S10 phosphorylation, supporting the conclusion that the S10D mutation primarily affects IRGQ engagement and basal organization of ATG8-positive structures rather than the inducible autophagy flux itself. This figure has been added to the manuscript as Figure 5B and replaces old Figure 5B and S5A, which were not normalised and missing the controls, as pointed out by both reviewers. We have updated the figure, legend, and manuscript text accordingly to reflect this clarification.

Figure R10: LC3-to-lysosome flux under stress is not changed when GABARAPL2 S10 phosphorylation is altered. Quantification of LC3-to-lysosome flux in GABARAPL2 WT, S10A, and S10D cells. Each point represents the normalized mean ratio per well and per biological repeat (three independent experiments, two wells per condition). Values were normalized to the untreated (UT) WT mean within each repeat. Conditions represent starvation (EBSS, 1–4 h), mitochondrial depolarization (40 μ M CCCP, 1–4 h), or lysosomal inhibition (200 nM BAF for 4 h; \pm EBSS or CCCP). Grouped one-sample t-tests compared combined EBSS (E1–E4), CCCP (C1–C4), and BAF (BAF, EBSS + BAF,

CCCP + BAF) groups to UT (= 1). **** $p < 0.0001$; *** $p < 0.001$; ** $p < 0.01$; * $p < 0.05$; n.s., not significant.

Reviewer #2 (Remarks to the Author):

In this study, the authors discover a specific functional role for GABARAPL2 in the modulation of IRGQ turnover through its phosphorylation by the kinase TBK1. The authors also discover the phospho site and show that phosphorylation of GABARAPL2 at S10 by TBK1 releases it from IRGQ and prevents the formation of autophagy initiation complexes. This was shown to have an effect on surface HLA turnover activity by IRGQ. Overall, this is a very interesting study. There are six ATG8 family members and while broad roles have been assigned to LC3 and GABARAP subfamily members, specific roles for ATG8 proteins are lacking, making the discovery here regarding GABARAPL2 one of the few reports to do so. In addition, TBK1 is largely a pro-selective autophagy kinase, but the authors show a negative regulation role for TBK1 in selective autophagy that is quite interesting. Given that TBK1 is strongly activated during selective autophagy of mitochondria and Salmonella, it would be beneficial for the authors to explore whether TBK1 activation to turn off IRGQ turnover is part of its role in enabling mitophagy or xenophagy, potentially by redistributing autophagy initiation complexes away from IRGQ and towards mitochondria or bacteria. Nevertheless, the study is of broad significance and interest, and the data are clear and largely convincing. Some specific suggestions are included below.

We thank the reviewer for this very positive and thoughtful assessment of our work. We appreciate the recognition that our study identifies a specific functional role for GABARAPL2 within the ATG8 family and provides mechanistic insight into how TBK1-mediated phosphorylation at GABARAPL2-S10 modulates IRGQ engagement, autophagy initiation complex formation, and downstream MHC-I turnover. We are also grateful for the reviewer's broader perspective, highlighting the novelty of defining a distinct function for one ATG8 paralog and for noting the unexpected, negative regulatory role of TBK1 in this selective autophagy context. Overall, we greatly value these constructive comments and have addressed the specific suggestions below to further strengthen the manuscript.

Major:

1. Figures 1 and 2: the Venus (BiCap) system binding between IRGQ and GABARAPL2 dependent on the LIR1 motif within IRGQ? This is a helpful control to clarify whether the Venus system binding experiments are largely driven between LIR mediated interactions or the Venus coming together. In addition, is the delivery to lysosomes of the Venus complex dependent on LIR1?

We thank the reviewer for this helpful suggestion. We agree that testing a LIR1-deficient IRGQ construct in the Venus assay would be a stringent additional control. While we did not perform the Venus experiment with an IRGQ LIR1 mutant (due to problems with cloning and plasmid generation), the available literature supports that split Venus fragments are engineered to be non-fluorescent and have low intrinsic self-association, and fluorescence is reconstituted only when the fused proteins are brought into proximity by a bona fide interaction (i.e., the reporter reports proximity

rather than driving binding).
https://pmc.ncbi.nlm.nih.gov/articles/PMC4417415/?utm_source=chatgpt.com,
https://pmc.ncbi.nlm.nih.gov/articles/PMC2829325/?utm_source=chatgpt.com,
<https://pubmed.ncbi.nlm.nih.gov/24947387/>

Importantly, we have validated in the manuscript that the functional outputs are LIR-dependent: IRGQ mutants in the N-terminal region that disrupt the relevant GABARAPL2-binding interface abolish the lipidation phenotype (Figure R7), and our structural modelling further supports a specific IRGQ–GABARAPL2 interface (as detailed in our response to Reviewer #1, Figures R2, R3, R4). On this basis, we interpret the Venus signal as reflecting LIR-mediated IRGQ–GABARAPL2 engagement, rather than Venus-driven association.

2. Figure 2D: The authors conclude that increased LC3B lipidation and increased LC3B foci correlates with increased autophagy. However, a block or decrease in autophagy flux can produce similar results. can the authors assess autophagy flux, perhaps through analysing p62 turnover, to assess whether autophagy flux is affected in cells with or without Venus system overexpression?.

We thank the reviewer for this important clarification. To directly assess autophagy flux and exclude the possibility that increased LC3B lipidation and foci reflect a block in autophagy, we analyzed p62/SQSTM1 turnover by immunoblotting. As expected, p62 levels were reduced upon EBSS treatment, confirming active autophagy flux. Importantly, upon Venus-based co-overexpression of IRGQ and GABARAPL2, p62 degradation was further enhanced, indicating increased autophagic flux rather than impaired turnover. These data demonstrate that the effects observed in Fig. 2D are not due to blocked autophagy. The new p62 turnover analysis has been added to the manuscript as Figure S2D.

Figure R11: SDS-PAGE and Western blot of HEK293T cell lysates with transfected V1-IRGQ and V2-GABARAPL2, untreated or treated with EBSS.

3. Figure 4C and S4G: Can the authors provide example images of the PLA experiments?

We have now included example images of Fig 4C as new Figure S6I.

Figure R12: Images of PLA signal (red) from endogenous GABARAP-L2 and endogenous IRGQ. HeLa endogenous HA-GABARAPL2 cells were infected with *Salmonella* (GFP) for 2 hours and fixed cells were probed with the Duolink in situ PLA assay. HA and IRGQ only antibodies were used as negative controls to determine the background.

4. After CCCP treatment, GABARAPL2 flux was considerably lower in WT and S10D cells compared to untreated conditions. Importantly, there was no difference between the two cell lines (Fig. S5B), confirming that TBK1 restricts the IRGQ-GABARAPL2 autophagy axis. Based on the above, what effect does the S10A mutant have on GABARAPL2 flux?

We thank the reviewer for raising this important point. We agree that the original presentation of Fig. S5B was not optimal (in particular, it was not appropriately normalized and lacked the relevant WT/S10A/S10D comparison controls). We have therefore removed Fig. S5B and replaced it with new Fig. S7A ,B, which include the missing controls and improved normalization. In the revised dataset, S10A phenocopies WT with respect to GABARAPL2 flux, whereas S10D shows the selective defect consistent with constitutive “phosphorylated-like” behavior. To strengthen this conclusion, we quantified autophagy flux using two independent readouts: microscopy-based analysis and FACS-based measurements, both of which support that S10A behaves like WT, while S10D reveals the functional consequence of enforced phosphorylation (new figure S7C). We now clarify in the manuscript that TBK1-mediated phosphorylation of GABARAPL2 at S10 acts as a conditional negative regulatory (“stop”) signal that terminates IRGQ–GABARAPL2 engagement and limits lysosomal trafficking once the pathway step is completed. In this framework, WT is predominantly unphosphorylated under basal conditions, and phosphorylation is expected to be transient and context-dependent, explaining why S10A often resembles WT. These additions have been incorporated into the Results/Discussion, and new Fig. S7A and S7B (replacing old Fig. S5B) now directly address the reviewer’s question by comparing GABARAPL2 WT, S10A, and S10D under matched conditions.

Figure R13: GABARAPL2-to-lysosome flux under stress is lost for the S10D phosphorylation mutant. Quantification of GABARAPL2-to-lysosome flux in cells expressing GABARAPL2 WT, S10A, or S10D. Each point represents the mean mCherry/GFP-mCherry puncta ratio per well and per biological replicate (four independent experiments, two wells per condition). Data were first normalized to the mean untreated (UT) signal averaged across WT, S10A, and S10D within each replicate, and then rescaled so that UT = 1 for all genotypes in the final plot. Conditions correspond to nutrient starvation (EBSS; 1–4 h), mitochondrial depolarization (40 μ M CCCP; 1–4 h), or lysosomal inhibition (200 nM BAF for 4 h; \pm EBSS or CCCP). One-sample Wilcoxon signed-rank with Holm p-value correction. **** $p < 0.0001$; *** $p < 0.001$; ** $p < 0.01$; * $p < 0.05$; n.s., not significant.

Figure R14: FACS analysis of GABARAPL2-to-lysosome flux in cells expressing Cherry-GFP GABARAPL2 WT, S10A, or S10D. Each point represents the mean mCherry/GFP-mCherry puncta ratio per well and per biological replicate. Data is presented as fold change and normalized to the mean untreated cell line (WT, S10A, or S10D, respectively). Conditions correspond to nutrient starvation (EBSS 6 h), mitochondrial depolarization (40 μ M CCCP 2 h), or lysosomal inhibition (200 nM BAF for 6 h + EBSS 6 h). $n = 3$.

5. In all experiments with CCCP treatment, are the cell lines also expressing Parkin? How was Parkin expression introduced? This seems to be missing from the methods.

We thank the reviewer for raising this point. None of the HeLa cell lines used in this study expresses Parkin. Parkin was only introduced in Figure R7 by overexpression. CCCP treatment alone was sufficient to induce TBK1 activation and subsequent

GABARAPL2 phosphorylation under our experimental conditions. We agree with the reviewer that ectopic Parkin expression would be required to achieve full mitophagy and potentially maximal TBK1 activation.

6. Figure 5B: Can the authors assess if other autophagy substrates/pathways are affected by S10D? For example, turnover of an ERphagy receptor, or p62 (with EBSS treatment), or alternatively if PINK1 Parkin mitophagy or Salmonella clearance is affected by the S10D mutation? This will help to clarify what effect S10D (TBK1 phosphorylation of GABARAPL2) might have on the activity of parallel selective autophagy pathways once IRGQ turnover is shut down.

We thank the reviewer for this insightful suggestion. To determine whether GABARAPL2-S10 phosphorylation broadly affects autophagy or selectively impacts IRGQ-dependent pathways, we assessed p62/SQSTM1 turnover as a readout of bulk and parallel selective autophagy fluxes. We quantified p62-to-lysosome flux in cells expressing GABARAPL2 WT, S10A, or S10D under starvation (EBSS), mitochondrial depolarization (CCCP), and lysosomal inhibition (BAF) conditions. p62 degradation was comparable between WT, S10A, and S10D cells, indicating that p62-dependent autophagy flux is independent of GABARAPL2-S10 phosphorylation. These data demonstrate that S10 phosphorylation does not impair bulk autophagy or selective autophagy pathways (consistent with new Fig 5B), but instead selectively affects IRGQ-dependent cargo turnover, exemplified here by MHC-I degradation (new Figure 5H). The p62 flux analysis and quantification have now been added to the manuscript as new Figure S7E, and the Results and Discussion sections have been updated accordingly.

0.001; ** $p < 0.01$; * $p < 0.05$; n.s., not significant.

Figure R15: p62-to-lysosome flux under stress conditions is independent of GABARAPL2 S10 phosphorylation.

Quantification of p62-to-lysosome flux in GABARAPL2 WT, S10A, and S10D cells. Each point represents the normalized mean ratio per well and per biological repeat (three independent experiments, two wells per condition). Values were normalized to the untreated (UT) mean within each genotype and repeat. Conditions represent starvation (EBSS, 1–4 h), mitochondrial depolarization (40 μ M CCCP, 1–4 h), or lysosomal inhibition (200 nM BAF for 4 h; \pm EBSS or CCCP). Grouped one-sample t-tests compared combined EBSS (E1–E4), CCCP (C1–C4), and BAF (BAF, EBSS + BAF, CCCP + BAF) groups to UT (= 1). **** $p < 0.0001$; *** $p <$

Minor

1. "TBK1 phosphorylates members of the LC3 family on several residues" Given that the authors go on to refer to GABARAPL2, it would be more accurate to state ATG8 family, LC3 is a subfamily of the ATG8s.

We thank the reviewer for pointing this out. The text has been changed accordingly.